# Search-R2: Enhancing Search-Integrated Reasoning via Actor-Refiner Collaboration

**Bowei He** [1 2 3 *] **Minda Hu** [4 3 *] **Zenan Xu** [3 *] **Hongru Wang** [5] **Licheng Zong** [4] **Yankai Chen** [1 2] **Chen Ma** [6] **Xue Liu** [1 2] **Pluto Zhou** [3] **Irwin King** [4]

## Abstract

Search-integrated reasoning enables language agents to transcend static parametric knowledge by actively querying external sources. However, training these agents via reinforcement learning is hindered by the *multi-scale credit assignment* problem: existing methods typically rely on sparse, trajectory-level rewards that fail to distinguish between high-quality reasoning and fortuitous guesses, leading to redundant or misleading search behaviors. To address this, we propose Search-R2, a novel Actor–Refiner collaboration framework that enhances reasoning through targeted intervention, with both components jointly optimized during training. Our approach decomposes the generation process into an Actor, which produces initial reasoning trajectories, and a Meta-Refiner, which selectively diagnoses and repairs flawed steps via a "cut-and-regenerate" mechanism. To provide fine-grained supervision, we introduce a hybrid reward design that couples outcome correctness with a dense process reward quantifying the information density of retrieved evidence. Theoretically, we formalize the Actor–Refiner interaction as a smoothed mixture policy, proving that selective correction yields strict performance gains over strong baselines. Extensive experiments across various general and multi-hop QA datasets demonstrate that Search-R2 consistently outperforms strong RAG and RL-based baselines across model scales, achieving superior reasoning accuracy with minimal overhead.

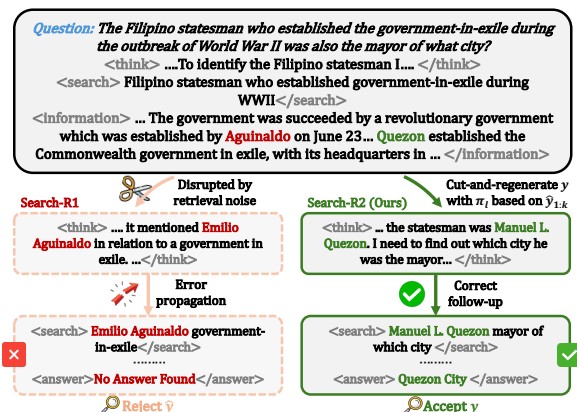

Figure 1. **Demonstration of Search-R1 and Search-R2.** While Search-R1 (Left) is disrupted by retrieval noise and falls into an error propagation loop, Search-R2 (Right) utilizes an Actor-Refiner collaboration. The Meta-Refiner identifies the deviation and applies a "cut-and-regenerate" mechanism to surgically repair the reasoning chain at the point of error, successfully redirecting focus from the incorrect entity (Aguinaldo) to the correct one (Quezon).

---

*Equal contribution [1]Mohamed bin Zayed University of Artificial Intelligence [2]McGill University [3]LLM Department, Tencent [4]The Chinese University of Hong Kong [5]The University of Edinburgh [6]City University of Hong Kong. Correspondence to: Hongru Wang <hrwang@ed.ac.uk>, Irwin King <king@cse.cuhk.edu.hk>, Pluto Zhou <plutozhou096@foxmail.com>.

*Proceedings of the 43rd International Conference on Machine Learning*, Seoul, South Korea. PMLR 306, 2026. Copyright 2026 by the author(s).

## 1. Introduction

Large language models are rapidly evolving from static knowledge repositories into dynamic, search-integrated agents that interact with external environments (Trivedi et al., 2023; Li et al., 2025b). By combining iterative reasoning with active retrieval, these agents tackle knowledge-intensive tasks such as open-domain and multi-hop question answering that were previously intractable due to limited parametric knowledge and hallucinations. This shift has been conceptually grounded by the recently proposed *Theory of Agent* (ToA) (Wang et al., 2025a), which reframes agents as tool-use decision makers operating under epistemic uncertainty and establishes a principled criterion: *external tools should be invoked only when epistemically necessary*, for distinguishing justified interaction from unnecessary delegation. Guided by this view, this field has turned to Reinforcement Learning (RL) to optimize these systems (Jin et al., 2025; Chen et al., 2026), grounding agent behavior in task-specific performance objectives rather than imitation of human demonstrations, and opening the door to training policies that calibrate when to reason internally and when

to retrieve externally.

However, training search-integrated agents with RL faces a key challenge: *multi-scale credit assignment*. In practice, agent behavior is a sequence of decisions, including query formulation, information filtering, and logical deduction, yet standard methods optimize policies with trajectory-level rewards such as final-answer correctness (Jin et al., 2025; Wang et al., 2025b). Since this outcome-only signal provides no supervision over intermediate reasoning or the timing and necessity of retrieval, it induces credit misattribution across both retrieval and reasoning decisions (Zhang et al., 2026). Consequently, efficient, logically coherent trajectories receive similar credit to trajectories that succeed only after redundant, costly, or poorly timed retrieval, reducing sample efficiency and yielding brittle reasoning chains. This limitation highlights a critical gap in current methodologies: *the inability to diagnose and repair error propagation*. As shown in Figure 1, a single irrelevant search query early in a trajectory can misguide the entire subsequent reasoning chain. Existing rejection sampling techniques (Ahn et al., 2024) are inefficient here, as they discard the entire trajectory rather than addressing the specific root cause of the deviation. To build robust agents, we must move beyond outcome-based filtering toward a paradigm that enforces both *global reasoning coherence* and *local search quality*.

To this end, we propose Search-R2, a novel Actor–Refiner collaboration framework designed to enhance search-integrated reasoning through targeted intervention. Unlike standard generation approaches, Search-R2 decomposes the reasoning process into two distinct roles: an Actor that generates initial reasoning trajectories with tool calls, and a Meta-Refiner that identifies localized failures, such as uninformative retrieval or logical gaps, and performs a "cut-and-regenerate" operation. This mechanism preserves valid reasoning prefixes while surgically repairing flawed suffixes, significantly enhancing learning efficiency. The Actor and Meta-Refiner are jointly optimized during training, enabling mutual feedback between trajectory generation and selective refinement. To further provide dense supervision, we introduce a hybrid reward that combines outcome correctness with a process reward that quantifies the density of evidence information. We theoretically prove that our Actor–Refiner interaction, which is modeled as a smoothed mixture policy, strictly exceeds the performance of baselines like rejection sampling under satisfiable conditions. Experiments on seven benchmarks show consistent gains of the proposed Search-R2 over strong RAG and RL-based baselines across model sizes ranging from 7B to 32B, with minimal overhead.

In summary, our contributions are as follows:

*1. Problem Identification*: We formalize the multi-scale credit assignment problem in search-integrated reasoning, highlighting the inadequacy of trajectory-level rewards for optimizing intermediate search behaviors.

*2. Framework*: We propose Search-R2, an Actor–Refiner framework that integrates step-level process rewards with a trajectory-level "cut-and-regenerate" refinement mechanism, and jointly optimizes both the Actor and the Refiner.

*3. Theoretical Analysis*: We characterize the Meta-Refiner as a mixture policy and derive the theoretical conditions under which selective correction guarantees performance improvement over baseline sampling.

*4. Empirical Success*: We demonstrate state-of-the-art performance on seven across different-size models, showing that Search-R2 improves both the accuracy of final answers and the quality of the underlying search process.

**Conflict of Interest Disclosure.** Some authors are affiliated with Tencent LLM Department. The evaluated method was developed by authors as part of this research. The base models used in experiments are publicly available Qwen models rather than models developed by the authors' employer.

## 2. Related Works

**Search-Integrated Reasoning.** Search-integrated language agents augment large language models with the ability to actively query external information sources during problem solving, enabling them to overcome the limitations of static parametric knowledge (Jin et al., 2025; Chen et al., 2026). Prior work has explored search-augmented reasoning for tasks such as multi-hop question answering (Sun et al., 2025; Wu et al., 2025), deep research (Team et al., 2025b; Hu et al., 2024), and web-based decision making (Zhou et al., 2024; Hu et al., 2025), demonstrating that iterative search can substantially improve factual accuracy and coverage. More recent approaches integrate search into reinforcement learning frameworks (Chen et al., 2026; Qian & Liu, 2026; Song et al., 2025), allowing agents to learn when and how to issue search queries based on task-level feedback. However, existing methods typically optimize search behavior only through delayed, trajectory-level rewards, without explicitly assessing the quality of individual search decisions (Wen et al., 2026). As a result, agents often issue redundant, mistimed, or weakly informative queries, especially in long-horizon interactions (Gao et al., 2025), where suboptimal search decisions compound over time and degrade both task performance and learning efficiency.

**Credit Assignment in Multi-Turn RL.** Learning effective policies for multi-turn decision making remains a central challenge in reinforcement learning and agent research due to sparse rewards and difficult credit assignment (Devidze et al., 2022; Wang & Ammanabrolu, 2025). This challenge is particularly pronounced in search-integrated agents, where intermediate decisions such as query formulation and timing are evaluated only through final task out-

comes (Zhang et al., 2026). Prior work has proposed dense reward shaping (Zeng et al., 2025; Zhang et al., 2025) and learned reward models (Zou et al., 2026), including Large Language Models (LLM)-based judges (Zha et al., 2026), to provide richer feedback signals. While these techniques improve optimization stability in some settings, they are most commonly applied to evaluate final responses or aggregate trajectory quality, leaving the quality of intermediate decisions underspecified. Consequently, policy optimization often suffers from low sampling efficiency, as many rollouts contain low-quality intermediate actions that contribute little to learning. These limitations motivate approaches that provide fine-grained supervision over intermediate decisions while remaining compatible with multi-turn optimization.

## 3. Methodology

We propose Search-R2, a novel Actor-Refiner collaboration framework designed to address the multi-scale credit assignment challenge. Rather than treating search-integrated reasoning as a monolithic generation task, our approach decouples the process into two distinct phases: an *Actor* generating initial reasoning chains, and a *Meta-Refiner* performing trajectory-level assessment and causal correction. This decomposition allows us to optimize both global reasoning coherence and local search quality simultaneously.

### 3.1. The Search-Integrated Reasoning Actor

The foundation of our system is an Actor policy, denoted as $\pi_l(\cdot|x)$, responsible for generating the initial reasoning trajectory $\hat{y}$. Given the search engine $\Lambda$, the $\pi_l$ is trained to invoke $\Lambda$ autonomously following a standard tool-use paradigm (Algorithm 2), to enable dynamic information acquisition. The model generates a chain of thought and, when necessary, emits a query within <search>...</search> tags. The system halts generation, executes the query against $\Lambda$, appends the top-$k$ results within <information>...</information> tags, and resumes generation. This cycle repeats until the model outputs the final answer or reaches a step limit.

To initialize $\pi_l$, we utilize a structural template (Table 1) that enforces a strict format: Reasoning → Search Call → Answer. This acts as a soft constraint, ensuring adherence to the system's operational logic without imposing content-specific biases.

*Table 1.* Template for Search-Integrated Reasoning following the implementation of Search-R1 (Jin et al., 2025).

| |
| --- |
| Answer the given question. You must conduct reasoning inside <think> and </think> first... if you lack knowledge, call search engine via <search> query </search>... return results in <information>... Final answer in <answer>... Question: question. |

### 3.2. The Meta-Refiner for Hierarchical Correction

A core premise of our work is that suboptimal search decisions often occur in intermediate steps and silently misguide subsequent reasoning. Standard rejection sampling is inefficient for repairing such cascading errors. To address this, we introduce the Meta-Refiner, which performs *targeted causal intervention* rather than blind regeneration. The Meta-Refiner shares the underlying LLM with the Actor but is steered by control prompts to perform two sub-objectives. Thus, we can avoid common multi-agent training challenges such as non-stationarity, inter-agent credit assignment, and inter-agent communication protocols.

*1) Discriminator for global coherence checking.* The Discriminator, denoted $\pi_d(\hat{y} \mid x) \in [0, 1]$, serves as a gate that enforces trajectory-level reasoning coherence. Given a reasoning trajectory $\hat{y}$, it estimates the probability that the reasoning remains globally coherent with the problem specified by $x$. We accept $\hat{y}$ when $\pi_d(\hat{y} \mid x) \geq \tau$; otherwise, we flag it for refinement. Accordingly, the acceptance probability is a Bernoulli distribution $\alpha(\hat{y}|x) = P(\pi_d(\hat{y} \mid x) \geq \tau)$.

*2) Trimmer for local error localization.* To address the issue of error propagation, the Trimmer $\pi_h(k|\hat{y}, x)$ identifies the specific search step $k + 1$ where the reasoning or search query first deviated (the "root cause"). The system preserves the valid prefix $\hat{y}_{1:k}$, truncates the flawed suffix, and regenerates a new suffix using the base policy $\pi_l$. This "cut-and-regenerate" strategy preserves valuable partial reasoning, significantly improving sample efficiency compared to discarding the entire trajectory.

Together, the discriminator and trimmer implement an iterative accept-or-repair procedure. For each candidate trajectory, the discriminator first decides whether it is globally coherent. If it is rejected, the trimmer localizes the earliest deviation and triggers cut-and-regenerate editing to produce a revised trajectory. This collaborative process induces a smoothed mixture policy $q(y \mid x)$, formalized in Algorithm 1. Repeating this procedure up to a budget $N_{\max}$ yields progressively improved trajectories and accumulates correction history, which strengthens the Meta-Refiner's ability to localize errors over time.

### 3.3. Hybrid Reward Modeling for Multi-Scale Supervision

To tackle the credit assignment issue where local search actions are conflated with global outcomes, we introduce a hybrid reward $R(y)$ that provides supervision at both scales.

*Global Outcome Reward.* We use Exact Match (EM) between the predicted answer $a_{\text{pred}}$ and ground truth $a_{\text{gold}}$: $r_{\text{outcome}}(y) = \mathbb{I}(a_{\text{pred}} = a_{\text{gold}})$. This ensures the final output satisfies the user's intent.

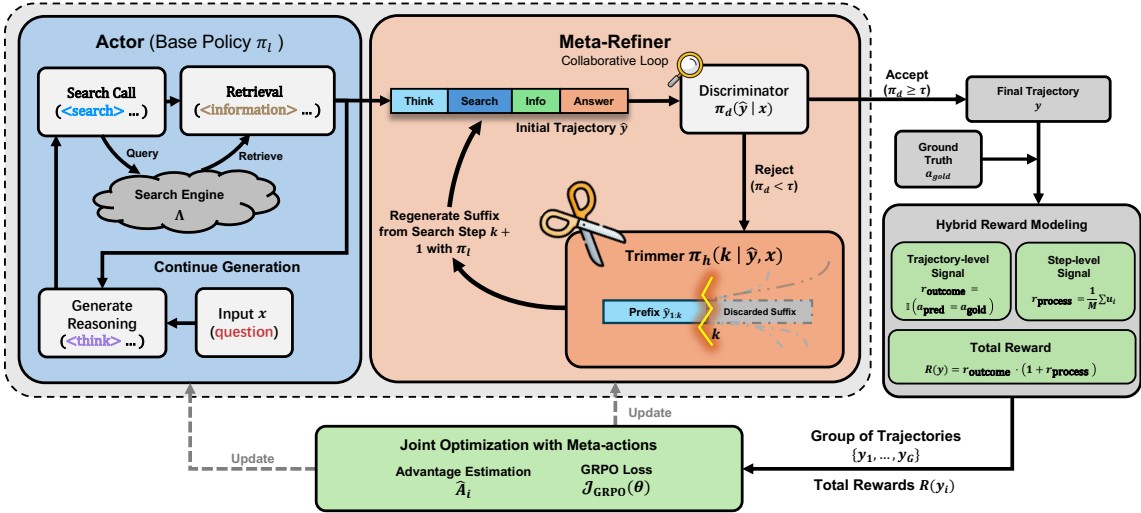

*Figure 2.* Overview of the Search-R2 framework. The Actor generates initial reasoning trajectories with search queries. The Meta-Refiner employs a Discriminator to detect errors and a Trimmer to identify the exact step of failure. Upon rejection, the trajectory is truncated and regenerated from the error point. The system is jointly optimized via GRPO using a hybrid reward.

---

**Algorithm 1** Meta-Refiner Execution Flow

---

1: **Input:** Context $x$, Policy $\pi_l$, Discriminator $\pi_d$, Trimmer $\pi_h$.
2: Generate initial trajectory $\hat{y} \sim \pi_l(\cdot|x)$
3: **while** $n < N_{max}$ **do**
4:   **if** $\pi_d(\hat{y}|x) \geq \tau$ **then**
5:     **return** $\hat{y}$ {Accept}
6:   **end if**
7:   Sample cut-point $k \sim \pi_h(\cdot|\hat{y}, x)$
8:   $y_{\text{prefix}} \leftarrow \hat{y}_{1:k}$
9:   Regenerate $y_{\text{suffix}} \sim \pi_l(\cdot|x, y_{\text{prefix}})$
10:   $\hat{y} \leftarrow [y_{\text{prefix}}, y_{\text{suffix}}]$
11:   $n \leftarrow n + 1$
12: **end while**
13: **return** $y = \hat{y}$

---

*Local Process Reward.* To distinguish between trajectories that are correct by chance versus those supported by high-quality evidence, we quantify the utility of retrieved context. For a set of retrieved chunks $C = \{c_1, \ldots, c_M\}$, an external judge evaluates the utility $u_i \in \{0, 1\}$ of each chunk. The process reward is the density of useful information: $r_{\text{process}}(y) = \frac{1}{M} \sum_{i=1}^{M} u_i$. Implementation specifics are outlined in Appendix M.

**Overall Reward.** To prevent reward hacking (maximizing retrieval without solving the task), the process reward is gated by outcome

$$R(y) = r_{\text{outcome}}(y) \cdot (1 + r_{\text{process}}(y)). \quad (1)$$

This formulation explicitly reinforces the principle that high-quality search is a necessary condition for robust reasoning.

### 3.4. Joint Optimizing the Actor and Meta-Refiner

We leverage Group Relative Policy Optimization (GRPO) to optimize the shared weight $\theta$ of Actor and Meta-Refiner

jointly (Shao et al., 2024). For each input $x$, we sample a group of $G$ trajectories $\{y_1, \ldots, y_G\}$ from the mixture distribution $q(\cdot|x)$. Crucially, we treat each $y_i$ as an augmented execution trace comprising both the reasoning path from $\pi_l$ and refinement actions sampled from the discriminator $\pi_d(y)$ and trimmer $\pi_h(k|\hat{y})$. The objective is to maximize:

$$
\begin{aligned}
\mathcal{L}_{\text{GRPO}}(\theta) &= \mathbb{E}_{x, \{y_i\}_{i=1}^{G} \sim q} \left[ \frac{1}{G} \sum_{i=1}^{G} \frac{1}{L_i} \sum_{t=1}^{L_i} L_t(y_i, \theta) \right] \\
L_t(y_i, \theta) &= \left[ r_t(\theta)\hat{A}_{i,t}, \text{clip}(r_t(\theta), 1 - \epsilon, 1 + \epsilon)\hat{A}_{i,t} \right] \\
&\quad - \beta \mathbb{D}_{\text{KL}}[\pi_l || \pi_{\text{ref}}],
\end{aligned}
\quad (2)
$$

where the advantage $\hat{A}_i$ is computed via group normalization of the hybrid rewards, and $r_t(\theta)$ denotes the probability ratio $\frac{\pi_\theta(a_t|s_t)}{\pi_{\theta_{old}}(a_t|s_t)}$, which measures the deviation of the current policy from the old policy. This allows the model to learn the optimal balance between generation and correction solely from the interaction outcome, effectively solving the multi-scale credit assignment problem end-to-end.

**Training vs. Inference Flow.** The Meta-Refiner is active *only during training rollouts*: during rollout, it explicitly evaluates trajectory coherence, identifies the earliest flawed step, and performs at most one cut-and-regenerate step per trajectory, and these meta-actions are included in the augmented GRPO trace. At deployment, by contrast, inference is a simplified, single-pass rollout: the Meta-Refiner is *not* invoked, and the benefits of refinement are implicitly encoded in the learned policy weights. As a result, Search-R2 introduces zero additional inference-time latency relative to the Search-R1 baseline. We illustrate the two flows in Figure 2 and Algorithm 1; an extended schematic is provided in Appendix E.

## 3.5. Mechanisms of Performance Gain

Unlike prior work that optimizes only the Actor, we jointly optimize both the Actor and the Meta-Refiner. To rigorously justify the necessity of optimizing the Meta-Refiner, as opposed to relying on static prompting or standard rejection sampling, we decompose the total expected sample reward improvement $\Delta J$, into three governing mechanisms. As formally derived in Appendix D, the net performance gain is not a byproduct of mere sampling volume, but strictly depends on the agent's ability to satisfy specific covariance conditions. We characterize the gain decomposition as:

$$\Delta J = \underbrace{\mathcal{A}_{\text{prec}}}_{\text{Selection Precision}} + \underbrace{\mathcal{V}_{\text{inter}}}_{\text{Intervention Volume}} \times \underbrace{\mathcal{S}_{\text{trim}}}_{\text{Trimming Skill}} . \quad (3)$$

We next describe each term in Eq. 3 and explain how it contributes to the overall improvement:

*Selection Precision $\mathcal{A}_{prec}$.* This term represents the system's capacity for global evaluation. Mathematically defined as $\text{Cov}_{\pi_l}(\alpha(y), R(y) - J_{\text{trim}}(y))$, it measures the alignment between the discriminator's acceptance probability and the trajectory's relative quality. A positive $\mathcal{A}_{\text{prec}}$ implies the discriminator successfully distinguishes which trajectories are worth preserving while exposing chains requiring correction. (e.g., those containing hallucinations or redundant steps) to the refinement process. By treating the entire interaction trace, such as reasoning, decision-to-accept, and decision-to-cut, as a single unified trajectory, GRPO naturally maximizes this covariance without requiring separate supervision signals for the Meta-Refiner.

*Trimming Skill $\mathcal{S}_{trim}$.* This term quantifies the effectiveness of the "cut-and-regenerate" mechanism. Defined as $\sum_k \text{Cov}(\pi_h(k|y), G_k(y))$, it measures the correlation between the selected cut-point $k$ and the expected gain $G_k(y)$ from regenerating at that specific step. Therefore, a positive $\mathcal{S}_{\text{trim}}$ indicates that the Trimmer precisely locates the specific low-quality search action that caused the reasoning collapse, such as a failed search query or a logic error, where the trajectory first deviated. This behavior is reinforced by propagating the outcome reward back to the specific cut-point selection $k$, encouraging the agent to target pivotal moments of failure.

*Intervention Volume $\mathcal{V}_{inter}$.* Defined as $1 - \mathbb{E}[\alpha(y)]$, this term represents the volume of trajectories subjected to correction. It acts as a multiplier in the Eq. 3. Even a highly skilled trimmer ($\mathcal{S}_{\text{trim}} > 0$) contributes little if the discriminator is overly conservative (accepting flawed answers, $\mathcal{V}_{\text{inter}} \to 0$). Conversely, if the discriminator flags valid answers ($\mathcal{V}_{\text{inter}} \to 1$) while the trimmer is unskilled, the computational budget is wasted. The system must find a balance between exploration and exploitation, ensuring that it neither overlooks errors nor wastes resources.

The joint optimization seeks an equilibrium where $\mathcal{V}_{\text{inter}}$ is sufficiently large to correct errors but constrained enough to preserve sample efficiency. Under joint optimization with meta-refiner, if the agent accepts a low-quality trajectory, the resulting low group-relative advantage penalizes the discriminator, directly driving $\mathcal{A}_{\text{prec}}$ upward.

**Summary of Success Conditions.** Unlike standard RAG or Rejection Sampling, which rely solely on the actor policy's generation probability, Search-R2 achieves a net positive gain ($\Delta J > 0$) for each rollout if and only if three conditions are met simultaneously. Formally, these correspond to $\mathcal{A}_{\text{prec}} > 0$, $\mathcal{S}_{\text{trim}} > 0$, and a calibrated $\mathcal{V}_{\text{inter}}$ that exposes sufficient samples for refinement without suppressing high-quality outputs. Furthermore, the Meta-Refiner supports iterative execution within $N_{max}$, where the posterior $q(\cdot|x)$ from iteration $t$ serves as the base policy for $t + 1$. The conditions for improvement remain valid in recursive settings.

# 4. Formalization

In this section, we present a theoretical framework for analyzing the mechanisms that drive the performance improvements of Search-R2. While the previous section detailed the algorithmic implementation of the Actor-Refiner collaboration, this section aims to mathematically quantify the specific contributions of the discrimination and refinement phases. We formalize the collaborative process as a smoothed mixture policy and derive a decomposition of the expected reward gain. We emphasize that this analysis is *descriptive* rather than prescriptive: it characterizes *when and why* the Actor–Refiner interaction yields strict gains over the base policy, via three interpretable conditions on selection precision, trimming skill, and intervention volume, rather than providing an absolute guarantee that any learned discriminator and trimmer will satisfy these conditions.

## 4.1. Performance Analysis

Our primary theoretical objective is to quantify the performance advantage of the Meta-Refiner over the base actor policy. We analyze the expected performance gain, $\Delta J = J_{meta} - J_{base}$, where $J_{base} = \mathbb{E}_{y \sim \pi_l}[R(y)]$ represents the standard actor's performance, and $J_{meta} = \mathbb{E}_{y \sim q}[R(y)]$ represents the performance under the Meta-Refiner distribution $q$. Analyzing this difference is crucial because it allows us to mathematically disentangle two sources of improvement, namely the *discriminative ability* to identify poor samples and the *trimming ability* to correct them.

**Proposition 4.1** (Performance Decomposition of Meta-Refiner). *Let the induced trajectory distribution $q(y \mid x)$ of the Meta-Refiner be formalized as a mixture policy:*

$$q(y \mid x) = \pi_l(y \mid x)\alpha(y) + \int_{\hat{y}} \pi_l(\hat{y} \mid x)(1 - \alpha(\hat{y}))T'(y \mid x, \hat{y}) \, d\hat{y}, \quad (4)$$

*where $\pi_l$ is the base policy, $\alpha(y) \in [0, 1]$ is the acceptance probability, and $T'(y \mid x, \hat{y})$ is the normalized transition*

*distribution of the trimmer for cutting and regenerating a rejected sample $\hat{y}$. Note that q is self-normalized (see Proof in Appendix B). The expected reward $J_{meta}$ decomposes relative to the base performance $J_{base}$ as:*

$$J_{meta} = J_{base}$$
$$+ \underbrace{\text{Cov}_{\pi_l}\left(a(y), R(y) - J_{trim}(y)\right)}_{\text{Selection Precision}} \quad (5)$$
$$+ \underbrace{(1 - Z_{acc})(\bar{J}_{trim} - J_{base})}_{\text{Correction Volume Gain}}.$$

*Here,* $\text{Cov}(X, Y) = \mathbb{E}[XY] - \mathbb{E}[X]\mathbb{E}[Y]$ *denotes the covariance.* $J_{trim}(\hat{y}) = \mathbb{E}_{y \sim T'(\cdot|\hat{y})}[R(y)]$ *is the expected reward after correcting* $\hat{y}$, $\bar{J}_{trim} = \mathbb{E}_{\pi_l}[J_{trim}(\hat{y})]$, *and* $Z_{acc} = \mathbb{E}_{\pi_l}[a(y)]$ *is the global acceptance rate.*

This derivation characterizes q as a smoothed mixture policy. The performance gain is driven by the discriminator's precision in identifying low-quality samples (*Selection Precision*) and the trimmer's ability to improve those samples (*Correction Volume Gain*).

### 4.2. Decomposing the Correction Volume Gain

We further analyze the term $\Delta J_{trim} = \bar{J}_{trim} - J_{base}$, which represents the performance improvement provided by the trimming strategy. We aim to decompose this gain into the *baseline gain* and the *attribution ability* of the trimmer.

**Preliminaries.** Let $\hat{y}$ be a draft sequence of length $T$ from $\pi_l$ rejected by the discriminator. We define the set of possible cut-points as $\mathcal{K} = \{1, \ldots, T\}$. Let $\pi_h(k|\hat{y})$ be the trimmer policy (probability of cutting at index $k + 1$) and let $V^{\pi_l}(\hat{y}_{1:k})$ be the value of regenerating the suffix from $k$.
**Proposition 4.2** (Decomposition of Trimming Strategy). *Let* $G_k(\hat{y}) = V^{\pi_l}(\hat{y}_{1:k}) - R(\hat{y})$ *denote the **regeneration gain** at step $k$. The total correction gain $\Delta J_{trim}$ decomposes into a covariance term representing the agent's skill and a mean term:*

$$\Delta J_{trim} = \underbrace{\sum_{k=1}^{T} \text{Cov}_{\hat{y}}(\pi_h(k|\hat{y}), G_k(\hat{y}))}_{\text{Trimming Skill}} + \underbrace{\bar{G}(\hat{y})}_{\text{Baseline Gain}}, \quad (6)$$

*where* $\bar{G}(\hat{y}) = \sum_k \mathbb{E}[\pi_h(k|\hat{y})]\mathbb{E}[G_k(\hat{y})]$ *denotes the baseline gain (see proof in Appendix C).*

This formulation isolates two drivers of performance:

- **Trimming Skill:** A positive covariance indicates that $\pi_h$ concentrates probability mass on cut-points $k + 1$ where the regeneration gain $G_k$ is highest. This measures the agent's ability to identify the "root cause" of a bad generation. A positive covariance implies that the trimmer possesses the capacity for concentrating probability mass on the critical turning points $k$ that yield the greatest regeneration gain ($G_k$) rather than performing random trimming.

- **Baseline Gain:** In high-dimensional reasoning tasks, arbitrarily truncating and regenerating a trajectory rarely improves the outcome (i.e., $\mathbb{E}[G_k(\hat{y})] \approx 0$ for random $k$). Consequently, $\bar{G} \approx 0$, implying that maximizing the correction gain $\Delta J_{\text{trim}}$ relies almost entirely on the trimmer's skill in selecting precise cut-points.

## 5. Experiments

### 5.1. Experiment Setup

**Datasets:** We evaluate search-integrated reasoning methods on two categories of datasets. For general question answering, we use NQ (Kwiatkowski et al., 2019), TriviaQA (Joshi et al., 2017), and PopQA (Mallen et al., 2022). For multi-hop question answering, we use HotpotQA (Yang et al., 2018), 2WikiMultiHopQA (Ho et al., 2020), Musique (Trivedi et al., 2022), and Bamboogle (Press et al., 2023). We train on the union of the NQ and HotpotQA training splits. Evaluation is performed on the validation or test splits of all seven datasets, which allows us to measure in-domain performance on the training distributions as well as out-of-domain generalization to held-out datasets.

**Methods:** We compare Search-R2 against three baseline families and a strong reference model. (i) *Inference without retrieval:* direct inference and Chain-of-Thought (CoT) reasoning (Wei et al., 2022). (ii) *Inference with retrieval:* Retrieval-Augmented Generation (RAG) (Lewis et al., 2020), IRCoT (Trivedi et al., 2023), and Search-o1 (Li et al., 2025b). (iii) *Fine-tuning based methods:* supervised fine-tuning (SFT) (Chung et al., 2024), RL-based fine-tuning without search (R1) (Guo et al., 2025), and rejection sampling with a search engine (Ahn et al., 2024). (iv) *Reference:* Search-R1 (Jin et al., 2025), the backbone of our approach. We run experiments on three model backbones spanning multiple generations and scales, namely Qwen2.5-32B, Qwen2.5-7B, and Qwen3-8B (Yang et al., 2024; 2025).

**Retriever:** We use E5 (Wang et al., 2022) as the retriever and the 2018 Wikipedia dump (Karpukhin et al., 2020) as the knowledge source. For fairness, we directly utilize the available index file provided by (Jin et al., 2025) and set the number of retrieved passages to 3.

**Implementation Details:** To ensure consistency with prior work (Jin et al., 2025), we use Exact Match (EM) as the evaluation metric and train all models with GRPO (Shao et al., 2024) for 300 steps. At each step, 512 prompts are randomly sampled, and $n = 5$ rollouts are generated for each prompt. Our training framework is based on the verl framework (Sheng et al., 2025) and sets the max assistant turns as 4. The max revision number per rollout is set as 1 by default. We use a learning rate of 1e-6 with a warmup ratio of 0.285. We provide more details in Appendix E.

*Table 2.* The main results on seven datasets. $^{\dagger}/^{\star}$ represents in-domain/out-of-domain datasets. All baselines except Search-R1 are conducted on the Qwen2.5-7B model. The best and second best performances are set as **bold** and underlined, respectively.

| Methods | General QA | | | | Multi-Hop QA | | | |
|---|---|---|---|---|---|---|---|---|
| | NQ$^{\dagger}$ | TriviaQA$^{\star}$ | PopQA$^{\star}$ | HotpotQA$^{\dagger}$ | 2WikiMultiHopQA$^{\star}$ | Musique$^{\star}$ | Bamboogle$^{\star}$ | Average |
| Direct Inference | 13.4 | 40.8 | 14.0 | 18.3 | 25.0 | 3.1 | 12.0 | 18.1 |
| CoT | 4.8 | 18.5 | 5.4 | 9.2 | 11.1 | 2.2 | 23.2 | 10.6 |
| IRCoT | 22.4 | 47.8 | 30.1 | 13.3 | 14.9 | 7.2 | 22.4 | 23.9 |
| Search-o1 | 15.1 | 44.3 | 13.1 | 18.7 | 17.6 | 5.8 | 29.6 | 20.6 |
| RAG | 34.9 | 58.5 | 39.2 | 29.9 | 23.5 | 5.8 | 20.8 | 30.4 |
| SFT | 31.8 | 35.4 | 12.1 | 21.7 | 25.9 | 6.6 | 11.2 | 20.7 |
| R1-base | 29.7 | 53.9 | 20.2 | 24.2 | 27.3 | 8.3 | 29.6 | 27.6 |
| R1-instruct | 27.0 | 53.7 | 19.9 | 23.7 | 29.2 | 7.2 | 29.3 | 27.1 |
| Rejection Sampling | 36.0 | 59.2 | 38.0 | 33.1 | 29.6 | 12.3 | 35.5 | 34.8 |
| Search-R1(Qwen2.5-7B) | 39.5 | 56.0 | 38.8 | 32.6 | 29.7 | 12.5 | 36.0 | 35.0 |
| Search-R1(Qwen3-8B) | 44.0 | 63.1 | 41.8 | 37.2 | 35.5 | 15.7 | 43.0 | 40.0 |
| Search-R1(Qwen2.5-32B) | 47.6 | 68.0 | 47.0 | 43.3 | 46.2 | 22.1 | 45.0 | 45.6 |
| Search-R2(Qwen2.5-7B) | 39.9 | 65.9 | 41.0 | 39.0 | 35.8 | 15.1 | 46.2 | 40.4 |
| Search-R2(Qwen3-8B) | 47.7 | 67.6 | 46.6 | 41.2 | 40.5 | 17.2 | 51.2 | 44.6 |
| Search-R2(Qwen2.5-32B) | **50.9** | **70.9** | **50.1** | **49.9** | **51.7** | **25.4** | **56.4** | **50.8** |

*Table 3.* Ablation results for Search-R2 on general and multi-hop question answering.

| Method | General QA | Multi-Hop QA | Average |
|---|---|---|---|
| **Qwen2.5-7B** | | | |
| Search-R1 | 41.7 | 26.1 | 35.0 |
| Search-R1 + Meta-Refiner | 45.3 | 30.4 | 38.9 |
| Search-R1 + Meta-Refiner + Process Reward | 45.6 | 31.6 | 39.6 |
| Search-R2 (Full Version) | **46.5** | **32.4** | **40.4** |
| **Qwen3-8B** | | | |
| Search-R1 | 46.5 | 31.4 | 40.0 |
| Search-R1 + Meta-Refiner | 49.4 | 35.3 | 43.4 |
| Search-R1 + Meta-Refiner + Process Reward | 49.9 | 36.1 | 44.0 |
| Search-R2 (Full Version) | **50.8** | **36.3** | **44.6** |
| **Qwen2.5-32B-Instruct** | | | |
| Search-R1 | 51.5 | 37.8 | 45.6 |
| Search-R1 + Meta-Refiner | 54.2 | 42.7 | 49.3 |
| Search-R1 + Meta-Refiner + Process Reward | 54.3 | 43.3 | 49.5 |
| Search-R2 (Full Version) | **55.5** | **44.5** | **50.8** |

## 5.2. Performance Comparison

Table 2 details the performance of Search-R2 against strong baselines on seven benchmarks. We observe that Search-R2 establishes a consistent performance lead. Notably, Search-R2 built on the Qwen2.5-7B backbone achieves a 16.1% EM gain over the Search-R1 rejection-sampling baseline, even when Search-R1 employs the stronger Qwen3-8B backbone. This confirms that the Actor-Refiner framework effectively compensates for reduced model scale by optimizing reasoning quality. When scaling the backbone from 7B to 32B, we observe a further performance gain, with average EM rising from 40.4 to 50.8. This consistent gain under model size scaling further highlights the effectiveness of our approach.

Moreover, the performance gains are more pronounced on complex reasoning tasks. For instance, Search-R2 achieves a 5.5-point improvement on 2WikiMultiHopQA and an 11.4-point improvement on Bamboogle (+25.3% relative

gain). These tasks typically require multi-step retrieval and reasoning, where early mistakes and noisy intermediate search results can cascade and derail the remaining trajectory. By using the Meta-Refiner to detect deviations and sample high-quality traces, Search-R2 mitigates such error propagation and yields larger gains across different benchmarks. Finally, to further verify that these gains stem from targeted refinement rather than additional computation, we compare Search-R2 against the Search-R1 baseline trained using a doubled rollout budget ($n = 10$). As reported in Appendix G, Search-R2 ($n = 5$, max revision $= 1$) still performs better, indicating that surgical correction is substantially more sample-efficient than brute-force sampling.

## 5.3. Ablation Study

To rigorously disentangle the sources of improvement in Search-R2, we perform a component-wise analysis by sequentially integrating the Meta-Refiner, Process Reward, and Joint Optimization modules into the Search-R1 baseline. For the intermediate configurations (Search-R1 + Meta-Refiner and + Process Reward), we optimize the policy solely on reasoning traces, excluding intervention refinement from the Meta-Refiner. As can be seen in Table 3, each module contributes positively to overall performance. Firstly, the integration of the Meta-Refiner drives the largest performance leap (+11.1% on Qwen2.5-7B), suggesting that the Meta-Refiner acts as a crucial scaffold for reasoning coherence. Secondly, integrating the process reward yields consistent performance gains by explicitly valuing high-information-density retrieval. It guides the Actor under sparse feedback in complex reasoning settings. Finally, the full Search-R2 setup with joint optimization achieves the highest accuracy. These results support our strategy:

*Table 4.* The hyperparameter sensitivity experiment results with increasing maximum revision times (from 1 to 4) for each initial rollout trajectory. We conduct these experiments on the Qwen2.5-32B-Instruct model. The best performance is set as **bold**.

| Max Revision | NQ | TriviaQA | PopQA | HotpotQA | 2WikiMultiHopQA | Musique | Bamboogle | Average |
|---|---|---|---|---|---|---|---|---|
| 1 | 50.8 | 69.4 | 49.0 | 47.6 | 49.4 | 24.2 | 54.4 | 49.3 |
| 2 | 50.9 | 71.0 | 50.6 | 48.7 | 50.7 | 25.5 | 54.4 | 50.2 |
| 3 | 51.4 | **71.2** | 50.4 | **49.3** | 51.4 | 25.7 | 54.4 | 50.6 |
| 4 | **51.6** | **71.2** | **50.8** | **49.3** | **51.6** | **26.0** | **55.6** | **50.9** |

unlike static methods, it enables the Actor and Meta-Refiner to co-adapt, allowing the policy to precisely localize errors and internalize the cut-and-regenerate mechanism for higher sample efficiency. Limited by the space, further details can be found in Appendix (Table 8).

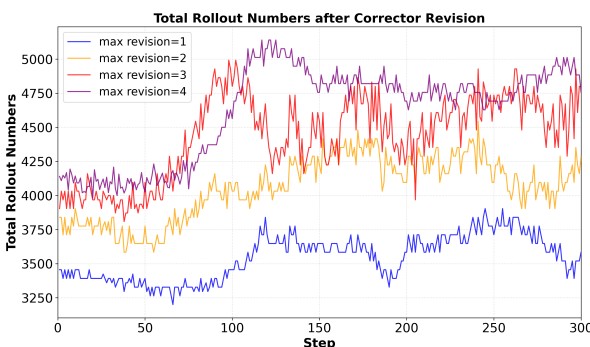

*Figure 3.* The total rollout numbers after revision (initial rollout numbers + refined rollout numbers) corresponding to different max revision time settings.

## 5.4. Sensitivity to the Maximum Revision Limit

We evaluate sensitivity to the maximum revision limit by varying the max revision value from 1 to 4 using Qwen2.5-32B as the backbone. In these experiments, we disable process reward modeling and joint optimization to focus on the effect of allowing additional revisions. As shown in Table 4, increasing max revision yields consistent gains. Notably, max revision = 4 reaches an average score of 50.9, essentially matching the fully optimized Search-R2 with a single revision (50.8). This comparison highlights an efficiency trade-off that our proposed joint optimization strategy can successfully distill the benefits of a larger revision budget into a more efficient policy that achieves comparable accuracy with one correction step.

We also observe rapidly diminishing gains as the revision limit increases. The absolute EM gain drops from 0.9 points when increasing revisions from 1 to 2 to 0.3 points from 3 to 4. This pattern suggests that early revisions primarily correct errors that are relatively easy to fix, such as retrieval noise or shallow hallucinations, whereas the remaining failures are less responsive to repeated refinement. Figure 3 corroborates this trend, showing that most trajectories trigger at most one revision. Given higher max revision limits, harder cases rarely activate further refinement. Consequently, we

set max revision = 1 as the default operating point, which captures most of the benefit at low revision cost.

*Table 5.* Average time cost for each training step (seconds/step).

| Model | Search-R1 | Search-R2 | Relative Change | $\frac{\Delta \text{EM} (\%)}{\Delta \text{Time} (\%)}$ |
|---|---|---|---|---|
| Qwen2.5-7B | 177.8 | 193.2 | + 8.66% | 1.78 |
| Qwen3-8B | 141.5 | 147.3 | + 4.10% | 2.80 |
| Qwen2.5-32B | 458.4 | 469.5 | + 2.43% | 4.69 |

## 5.5. Efficiency Analysis

We now examine whether the Search-R2 pipeline introduces substantial computational overhead in practice. Surprisingly, Table 5 shows that Search-R2 increases training time by only 5.06% on average relative to the Search-R1 baseline. This modest overhead is largely due to the cut-and-regenerate mechanism, which preserves valid prefixes rather than discarding entire trajectories, thereby reducing wasted computation. Moreover, the relative overhead decreases with model scale and drops to 2.43% for the 32B model, suggesting that the marginal refinement cost becomes less significant as distributed training overhead grows. At inference time, Search-R2 introduces no additional latency because the Meta-Refiner is decoupled at deployment.

To quantify training cost-effectiveness, we report the ratio $\Delta \text{EM}(\%)/\Delta \text{Time}(\%)$, which measures accuracy improvement per unit increase in training time. As shown in Table 2, this ratio exceeds 1 for all models, indicating that accuracy gains consistently outpace the added compute. Moreover, the ratio further improves with scale, increasing from 1.78 at 7B to 4.69 at 32B, which suggests that Search-R2 becomes more cost-effective for larger backbones.

**Memory Overhead.** Because the discriminator and trimmer share parameters with the Actor (no extra standalone models are loaded), Search-R2 also introduces minimal memory overhead at training time. On our $8\times$H20-96GB nodes, peak per-GPU memory rises from $\sim$60 GB to $\sim$62 GB ($+2$ GB, $< 3.5\%$) for the Qwen2.5-7B backbone under GRPO, and from $\sim$86 GB to $\sim$88 GB ($+2$ GB, $<2.5\%$) for the Qwen2.5-32B backbone. The marginal $\sim$2 GB stems from lightweight bookkeeping for refinement decisions during rollout; the dominant memory footprint continues to come from the long-context rollouts (up to 15k tokens) and multi-trajectory sampling ($n\!=\!5$) that are standard in multi-

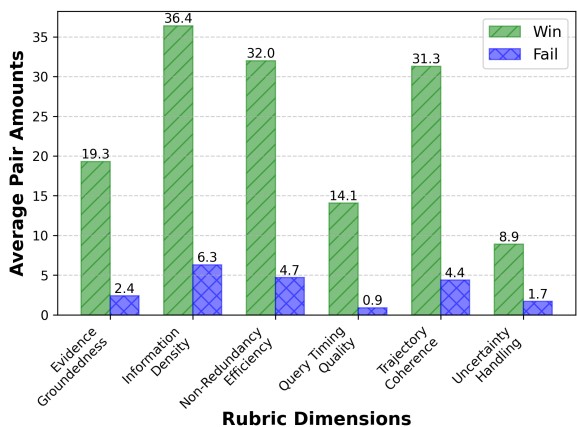

*Figure 4.* Average counts of Search-R2 winning and failing against Search-R1 across all seven datasets for each rubric.

turn RL. The method fits comfortably within a 96 GB budget without requiring tensor parallelism for the 7B/8B models, and remains tractable at 32B. At inference time, as discussed in Section 3.4, the Meta-Refiner is not invoked and therefore Search-R2 incurs *no* additional memory or latency overhead relative to Search-R1.

### 5.6. Trajectory Quality Comparison

To better understand trajectory quality, we compare Search-R2 against Search-R1 using `GPT-5.1` as an automated judge. The evaluation covers six dimensions: *evidence groundedness*, *information density*, *non-redundancy efficiency*, *query timing quality*, *trajectory coherence*, and *uncertainty handling*. For each of the seven test datasets, we randomly sample 100 paired trajectories, evaluating Search-R1 and Search-R2 on the same prompt, for a total of 700 pairs. The judge assigns each trajectory an independent three-level score, with 0 indicating poor quality, 1 acceptable, and 2 strong. We then compare the paired scores and record a win when Search-R2 scores higher, a fail when it scores lower, and a tie otherwise[1]. As shown in Figure 4, Search-R2 outperforms Search-R1 across all dimensions, indicating more grounded, efficient, and coherent search and reasoning behavior. Detailed rubrics, full results, and evaluation prompts are provided in Appendix K.

**Robustness to the Choice of Judge.** To rule out the possibility that the observed quality gains are an artifact of any single LLM judge, we re-evaluated 100 randomly sampled trajectory pairs per dataset using two additional evaluators: an independent LLM judge (`Claude Opus 4.6`) and a panel of five human NLP experts. Aggregating across all seven datasets, Search-R2 is preferred over Search-R1 with win/fail rates of $23.7\%/3.4\%$ (`GPT-5.1`), $28.7\%/4.1\%$ (`Claude Opus 4.6`), and $21.4\%/4.5\%$ (human experts), i.e., a $5-7\times$ margin under every eval-

uator. Inter-annotator agreement is substantial under Fleiss' $\kappa$ (human: $0.68$; cross-LLM: $0.71$). The per-dataset breakdown and human-evaluation protocol are reported in Appendix K.4.

## 6. Conclusions and Future Works

In this work, we introduced Search-R2, a search-integrated reasoning framework designed to mitigate the LLM fragility when facing retrieval noise. Experiments show that while standard approaches like Search-R1 are susceptible to error propagation loops caused by misleading initial context, Search-R2's Actor-Refiner collaboration with joint optimization effectively interrupts these failures. By employing a dynamic "cut-and-regenerate" mechanism, Search-R2 enables models to correct reasoning trajectories in real-time. These findings highlight the critical importance of integrating active refinement into search-integrated reasoning, offering a path toward more reliable agent behavior. Looking forward, several directions remain open. First, the process reward currently relies on an external LLM judge with access to the ground-truth answer; designing fully answer-free reward signals based on query–document relevance, evidence coverage, or content-reasoning consistency is a promising next step. Second, our evaluation is conducted under a controlled retrieval setting, and extending Search-R2 to noisier, long-horizon, open-web environments with richer toolsets (e.g., browsers, code execution, file management) would better stress-test the framework. Finally, the Actor-Refiner decomposition is orthogonal to the choice of RL optimizer; integrating it into broader agentic pipelines and combining it with other techniques such as larger rollout budgets or alternative RL algorithms could further amplify its benefits.

## Impact Statement

This work presents an Actor-Refiner collaboration framework designed to enhance the reliability and efficiency of language agents in multi-turn search-integrated reasoning. By implementing a mechanism for the selective correction of low-quality trajectories and incorporating retrieval-aware process rewards, our approach mitigates unnecessary computational overhead and reduces erroneous search queries. This results in superior reasoning performance with only a marginal increase in training costs. However, as with many retrieval-augmented systems, our method relies on external information sources. This dependence introduces potential risks regarding the propagation of bias or misinformation found in retrieved documents, underscoring the necessity for robust content filtering and rigorous evaluation protocols in real-world deployments. We further encourage practitioners deploying such systems to pair them with transparent provenance tracking, so that end users can inspect which retrieved sources support each model response.

---

[1] We omit ties in Figure 4 to improve readability.

## Acknowledgements

The research presented in this paper was primarily supported by the MBZUAI Base Fund. In addition, it was partially supported by the Research Grants Council of the Hong Kong Special Administrative Region, China (CUHK 2300246, RGC C1043-24G), (CUHK 14203425, RGC GRF 2151317), and CUHK 7010870.

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

# Contents

# A. Notation Table

Table 6 summarizes the mathematical notations and symbols used throughout the formalization and analysis of the Search-R2 framework.

*Table 6.* Summary of Notations and Symbols

| Symbol | Description |
|---|---|
| *Policies and Models* | |
| $\pi_l(y\|x)$ | The **Base Policy (Actor)** responsible for generating reasoning trajectories and search queries. |
| $\pi_d(\hat{y}\|x)$ | The **Discriminator (part of Meta-Refiner)** that estimates the probability of a trajectory's global coherence. |
| $\pi_h(k\|\hat{y}, x)$ | The **Trimmer (part of Meta-Refiner)** that identifies the specific step $k$ (cut-point) where an error occurred. |
| $q(y\|x)$ | The **smoothed mixture policy** induced by the interaction between the Actor and the Meta-Refiner realized by Algorithm 1. |
| $\Lambda$ | The external search engine used for retrieval. |
| *Trajectory and Search* | |
| $x$ | The input context or question. |
| $\hat{y}$ | The initial reasoning trajectory generated by the Actor $\pi_l$. |
| $y$ | The final trajectory output after potential refinement from $q(y\|x)$. |
| $a_{\text{pred}}$ | The predicted final answer extracted from the trajectory. |
| $a_{\text{gold}}$ | The ground truth answer. |
| $k$ | The index of a step in the trajectory (specifically used as the cut-point). |
| $\hat{y}_{1:k}$ | The valid prefix of trajectory $\hat{y}$ up to step $k+1$. |
| $C$ | A set of retrieved chunks $\{c_1, \ldots, c_M\}$. |
| *Rewards and Optimization* | |
| $R(y)$ | The total hybrid reward described in Section 3.3, combining outcome and process signals. |
| $r_{\text{outcome}}(y)$ | The outcome reward (Binary Exact Match) indicating if $a_{\text{pred}} = a_{\text{gold}}$. |
| $r_{\text{process}}(y)$ | The process reward quantifying the information density of retrieved evidence. |
| $\mathcal{L}_{\text{GRPO}}(\theta)$ | The objective function for Group Relative Policy Optimization. |
| $\hat{A}_t$ | The advantage estimate computed via group normalization. |
| *Theoretical Analysis* | |
| $\text{Cov}(X, Y)$ | The covariance between variables $X$ and $Y$, defined as $\mathbb{E}[XY] - \mathbb{E}[X]\mathbb{E}[Y]$. |
| $\alpha(\hat{y}\|x)$ | The Discriminator $\pi_d$'s acceptance probability of a trajectory, defined as $P(\pi_d(\hat{y}\|x) \geq \tau)$. |
| $\tau$ | The predefined threshold for the Discriminator $\pi_d$ to accept a trajectory. |
| $Z_{\text{acc}}$ | The global acceptance rate, defined as $\mathbb{E}_{\pi_l}[\alpha(y)]$. |
| $J_{\text{base}}$ | The expected performance of the base policy: $\mathbb{E}_{y \sim \pi_l}[R(y)]$. |
| $J_{\text{meta}}$ | The expected performance of the meta-refiner policy: $\mathbb{E}_{y \sim q}[R(y)]$. |
| $\Delta J$ | The net performance gain: $J_{\text{meta}} - J_{\text{base}}$. |
| $\mathcal{A}_{\text{prec}}$ | *Selection Precision*: Covariance measuring the Discriminator's ability to identify low-quality samples. |
| $\mathcal{S}_{\text{trim}}$ | *Trimming Skill*: Covariance measuring the Trimmer's ability to locate the root cause of errors. |
| $\mathcal{V}_{\text{inter}}$ | *Intervention Volume*: The probability mass allocated to the trimming process $(1 - Z_{\text{acc}})$. |
| $J_{\text{trim}}(\hat{y})$ | The expected reward after correcting a specific rejected trajectory $\hat{y}$. |
| $G_k(\hat{y})$ | The regeneration gain at step $k$: $V^{\pi_l}(\hat{y}_{1:k}) - R(\hat{y})$. |
| $V^{\pi_l}(\hat{y}_{1:k})$ | The value of regenerating the suffix starting from step $k$ using the base policy $\pi_l$. |

## B. Proof for Performance Decomposition of Meta-Refiner

*Proof.* **1. Normalization Check.** We first verify that $q(y|x)$ integrates to 1.

$$\int q(y|x)dy = \int \pi_l(y)\alpha(y)\, dy + \int \left[ \int \pi_l(\hat{y})(1-\alpha(\hat{y}))T'(y \mid \hat{y})\, d\hat{y} \right] dy$$

$$= \mathbb{E}_{\pi_l}[\alpha(y)] + \int \pi_l(\hat{y})(1-\alpha(\hat{y}))\underbrace{\left[ \int T'(y \mid \hat{y})\, dy \right]}_{=1} d\hat{y}$$

$$= Z_{acc} + \mathbb{E}_{\pi_l}[1-\alpha(\hat{y})]$$

$$= Z_{acc} + (1 - Z_{acc}) = 1. \tag{7}$$

**2. Expected Reward Derivation.** The expected reward $J_{meta}$ is the integral of $R(y)$ over the mixture components:

$$J_{meta} = \underbrace{\int R(y)\pi_l(y)\alpha(y)\, dy}_{\text{Term A (Accepted)}} + \underbrace{\int R(y)\left[ \int \pi_l(\hat{y})\bar{\alpha}(\hat{y})T'(y \mid \hat{y})\, d\hat{y} \right] dy}_{\text{Term B (Rejected)}}, \tag{8}$$

where $\bar{\alpha}(\hat{y}) = 1 - a(\hat{y})$.

*Analyzing Term A:* Using the covariance identity $\mathbb{E}[XY] = \mathbb{E}[X]\mathbb{E}[Y] + \text{Cov}(X,Y)$:

$$A = \mathbb{E}_{y \sim \pi_l}[R(y)\alpha(y)] = J_{base}Z_{acc} + \text{Cov}_{\pi_l}(\alpha, R).$$

*Analyzing Term B:* By Fubini's Theorem, we swap the order of integration:

$$B = \int \pi_l(\hat{y})(1-\alpha(\hat{y}))\left[ \int R(y)T'(y \mid \hat{y})\, dy \right] d\hat{y}$$

$$= \int \pi_l(\hat{y})(1-\alpha(\hat{y}))J_{\text{trim}}(\hat{y})\, d\hat{y}$$

$$= \mathbb{E}_{\hat{y} \sim \pi_l}\left[ (1-\alpha(\hat{y}))J_{\text{trim}}(\hat{y}) \right]. \tag{9}$$

Let $\bar{J}_{trim} = \mathbb{E}_{\pi_l}[J_{trim}(\hat{y})]$. Applying the covariance identity again:

$$B = (1 - Z_{acc})\bar{J}_{trim} - \text{Cov}_{\pi_l}(\alpha, J_{trim}). \tag{10}$$

*Synthesis:* Combining A and B, and grouping the covariance terms:

$$J_{\text{meta}} = \left[ J_{base}Z_{acc} + (1 - Z_{acc})\bar{J}_{trim} \right] + \left[ \text{Cov}_{\pi_l}(\alpha, R) - \text{Cov}_{\pi_l}(\alpha, J_{trim}) \right]$$

$$= \left[ J_{base}Z_{acc} + (1 - Z_{acc})\bar{J}_{trim} \right] + \text{Cov}_{\pi_l}(\alpha, R - J_{trim}).$$

Subtracting $J_{base}$ from both sides yields the final gain:

$$\Delta J = \text{Cov}_{\pi_l}(\alpha, R - J_{trim}) + (1 - Z_{acc})(\bar{J}_{trim} - J_{base}).$$

$\square$

## C. Proof for Decomposition of Trimming Strategy

*Proof.* The total expected gain is the difference between the expected return after trimming and the baseline:

$$\Delta J_{\text{trim}} = \mathbb{E}_{\hat{y}}\left[ \sum_{k=1}^{T} \pi_h(k|\hat{y})V^{\pi_l}(\hat{y}_{1:k}) \right] - \mathbb{E}_{\hat{y}}[R(\hat{y})]$$

$$= \mathbb{E}_{\hat{y}}\left[ \sum_{k=1}^{T} \pi_h(k|\hat{y})\left( V^{\pi_l}(\hat{y}_{1:k}) - R(\hat{y}) \right) \right]$$

$$= \sum_{k=1}^{T} \mathbb{E}_{\hat{y}}\left[ \pi_h(k|\hat{y})G_k(\hat{y}) \right]. \tag{11}$$

Applying the covariance identity $\mathbb{E}[XY] = \mathrm{Cov}(X, Y) + \mathbb{E}[X]\mathbb{E}[Y]$ to each term in the summation yields the proposition.

$\square$

# D. Drivers of Performance Gain

Building upon Propositions 4.1 and 4.2, we decompose the total system improvement, $\Delta J$, into three governing factors. These components isolate the specific contributions of the discriminator's judgment, the Meta-Refiner's localization capability, and the overall frequency of intervention.

**Definition D.1** (Selection Precision). Let $\mathcal{A}_{\mathrm{prec}}$ quantify the covariance between the acceptance probability $\alpha(y)$ and the sample's relative advantage (current reward minus potential correction value):

$$\mathcal{A}_{\mathrm{prec}} \triangleq \mathrm{Cov}_{\pi_l}\left(\alpha(y),\ R(y) - J_{trim}(y)\right). \tag{12}$$

A positive $\mathcal{A}_{\mathrm{prec}}$ indicates that the discriminator functions as an effective filter, preferentially preserving samples where the existing reward $R(y)$ outweighs the expected value of a correction $J_{trim}(y)$.

**Definition D.2** (Trimming Skill). Let $\mathcal{S}_{\mathrm{trim}}$ quantify the alignment between the cut-point policy $\pi_h$ and the regeneration gain $G_k(\hat{y})$ across all possible cut-points $k$:

$$\mathcal{S}_{\mathrm{trim}} \triangleq \sum_{k=1}^{T} \mathrm{Cov}_{\pi_l}\left(\pi_h(k \mid \hat{y}),\ G_k(\hat{y})\right). \tag{13}$$

A positive $\mathcal{S}_{\mathrm{trim}}$ implies the Meta-Refiner correctly identifies cut-points $k$ that yield higher regeneration gains.

**Definition D.3** (Intervention Volume). Let $\mathcal{V}_{\mathrm{inter}}$ represent the total probability mass allocated to the trimming process (the rejection rate):

$$\mathcal{V}_{\mathrm{inter}} \triangleq 1 - Z_{acc} = \mathbb{E}_{\pi_l}[1 - \alpha(y)]. \tag{14}$$

This term dictates the magnitude of the opportunity space available for the Trimmer to act.

Substituting these definitions into the total gain equation yields the following decomposition:

$$\Delta J = \mathcal{A}_{\mathrm{prec}} + \mathcal{V}_{\mathrm{inter}} \cdot (\mathcal{S}_{\mathrm{trim}} + \underset{\approx 0}{\bar{G}}). \tag{15}$$

We leverage GRPO with meta-actions to jointly optimize the Actor and the Meta-Refiner. By treating each trajectory $y_i$ as an augmented execution trace, comprising both the reasoning tokens from $\pi_l$ and the meta-actions sampled from the Discriminator $\pi_d(\hat{y})$ and Trimmer $\pi_h(k|\hat{y})$. GRPO inherently maximizes $\Delta J$. This formulation ensures that the policy gradient updates align with the maximization of $\mathcal{A}_{\mathrm{prec}}$ and $\mathcal{S}_{\mathrm{trim}}$.

# E. Supplementary Implementation Details

**Training vs. Inference Flow.** During training, every prompt produces $n = 5$ initial rollouts from $\pi_l$; for each rollout, the Meta-Refiner is invoked once, may either accept the trajectory or trigger a cut-and-regenerate from the localized error step, and *all* of these meta-decisions are recorded as part of the augmented GRPO trace for policy-gradient updates. During deployment, by contrast, the Meta-Refiner is *not* invoked at all; inference reduces to the standard Search-R1-style multi-turn rollout (Algorithm 2), and the refinement skill is encoded only implicitly in the learned weights. This design choice means Search-R2's training-time gains do *not* come at the cost of inference-time latency or memory.

**Hardware.** All experiments were conducted on multiple 8-node GPU clusters. Each node features dual-socket AMD EPYC 9K84 processors, providing a total of 192 physical cores and 384 threads per node, organized into two NUMA nodes. Storage infrastructure includes a 480 GB SATA SSD for the OS and environment, alongside two enterprise-grade 7.68 TB NVMe SSDs for high-throughput local data caching. Nodes are linked via a high-speed interconnect and share a distributed file system for dataset storage and checkpoint synchronization.

**Configurations.** The model is trained on a unified search-integrated reasoning dataset stored in Parquet format. *Data & Rollout:* We set the maximum prompt and response lengths to 4096 and 3000 tokens, respectively. To prevent information loss, truncation is disabled; prompts exceeding the limit are filtered out. We utilize SGLang as the rollout engine to

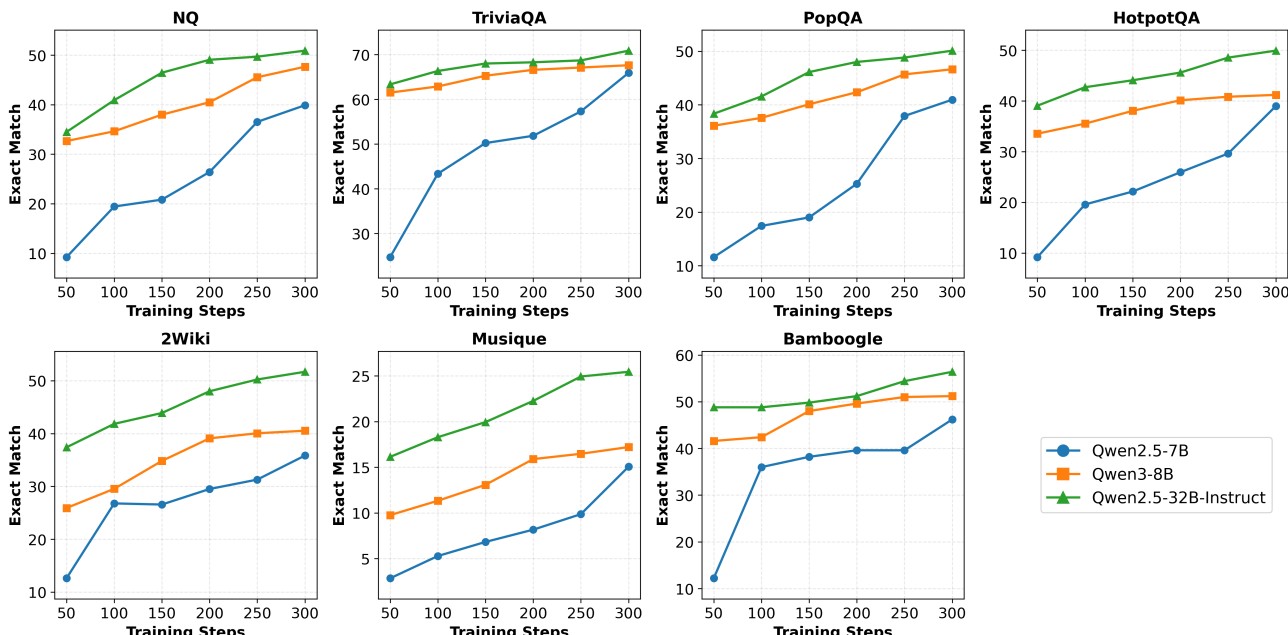

*Figure 5.* Detailed training dynamics of Search-R2 with different base models across all seven datasets.

facilitate efficient multi-turn generation with tool calls, maintaining the raw chat format. Each prompt samples $n = 5$ rollout trajectories per GRPO step, with a maximum of 4 assistant turns per trajectory. The context length during rollout is capped at 15,000 tokens to accommodate interleaved reasoning and retrieved evidence. For validation, we employ greedy decoding (sampling disabled). *Optimization:* The Actor is trained via PPO-style updates using GRPO advantages. We utilize a learning rate of 1e-6 with a warmup ratio of 0.285. The global PPO mini-batch size is 512, with a per-GPU micro-batch size of 4. To stabilize training, we apply a low-variance KL penalty (coefficient 0.001) rather than incorporating it into the reward; entropy regularization is disabled. Training utilizes Fully Sharded Data Parallel (FSDP) with full state offloading. Tensor model parallelism is set to 8 for the 32B model and 2 for the 7B/8B models. *Meta-Refiner:* The Meta-Refiner functions as an internal agent sharing weights with the Actor but utilizing distinct prompts. It is trained jointly with the Actor and remains active during rollout, performing at most one revision per trajectory. Intervention decisions are determined by comparing log-probabilities of candidate actions (revision vs. no-revision); a revision is triggered only if its log-probability exceeds that of the no-revision decision (margin $\geq 0.0$).

**Resource Links.** We provide the necessary resource links of models, retrievers, and software, to help reproduce our implementation and experiments as follows: *Models:* Qwen2.5-32B-Instruct [2], Qwen2.5-7B [3], Qwen3-8B [4], and DeepSeek-R1-Distill-Qwen-7B [5]; *Retriever:* E5 [6], 2018 Wikipedia dump [7], and index file [8]; *Softwares:* verl [9], FSDP [10], and SGlang [11].

# F. Training Dynamics

To investigate the training dynamics of our agentic RL framework, Figure 5 visualizes EM scores across the seven experimental datasets, plotted from 0 to 300 steps at 50-step intervals. We observe consistent trends across all three models and datasets, with performance converging as training approaches 300 steps. Extending training beyond this point yields

---

[2] https://huggingface.co/Qwen/Qwen2.5-32B-Instruct
[3] https://huggingface.co/Qwen/Qwen2.5-7B
[4] https://huggingface.co/Qwen/Qwen3-8B
[5] https://huggingface.co/deepseek-ai/DeepSeek-R1-Distill-Qwen-7B
[6] https://huggingface.co/intfloat/e5-base-v2
[7] https://huggingface.co/datasets/PeterJinGo/wiki-18-corpus
[8] https://huggingface.co/datasets/PeterJinGo/wiki-18-e5-index
[9] https://github.com/volcengine/verl/tree/main
[10] https://docs.pytorch.org/docs/stable/fsdp.html
[11] https://github.com/sgl-project/sglang

*Table 7.* Performance comparison between Search-R2 (initial rollout number as 5 per prompt and max revision as 1) and Search-R1 with double rollout numbers (10 per prompt instead of default 5 per prompt). Here, Qwen2.5-32B-Instruct is taken as the base model.

| Training Steps | NQ | TriviaQA | PopQA | HotpotQA | 2WikiMultiHopQA | Musique | Bamboogle | Average |
|---|---|---|---|---|---|---|---|---|
| **Search-R1 ($n = 10$)** | | | | | | | | |
| 50 | 33.9 | 63.3 | 38.2 | 38.7 | 36.6 | 15.7 | 46.4 | 39.0 |
| 100 | 38.3 | 65.3 | 40.3 | 41.4 | 40.2 | 17.4 | 45.6 | 41.2 |
| 150 | 43.8 | 67.6 | 45.2 | 43.7 | 43.9 | 18.9 | 49.6 | 44.7 |
| 200 | 46.6 | 68.0 | 47.3 | 44.3 | 44.5 | 21.1 | 48.8 | 45.8 |
| 250 | 49.0 | 68.3 | 49.1 | 45.2 | 46.6 | 23.2 | 50.4 | 47.4 |
| 300 | 49.7 | 68.6 | 49.1 | 45.9 | 47.8 | 24.0 | 49.6 | 47.8 |
| **Search-R2 ($n = 5$, max revision = 1)** | | | | | | | | |
| 50 | 34.5 | 63.3 | 38.3 | 39.1 | 37.4 | 16.1 | 48.8 | 39.7 |
| 100 | 40.9 | 66.3 | 41.6 | 42.7 | 41.8 | 18.3 | 48.8 | 42.9 |
| 150 | 46.5 | 68.0 | 46.1 | 44.1 | 43.9 | 19.9 | 49.8 | 45.5 |
| 200 | 49.1 | 68.3 | 48.0 | 45.6 | 48.0 | 22.3 | 51.2 | 47.5 |
| 250 | 49.7 | 68.7 | 48.8 | 48.6 | 50.2 | 24.9 | 54.4 | 49.3 |
| 300 | 50.9 | 70.9 | 50.1 | 49.9 | 51.7 | 25.4 | 56.4 | 50.8 |

negligible performance gains and increases the risk of model collapse due to instabilities such as train–inference mismatch and automatic mixed-precision overflow—challenges inherent to the current RL training infrastructure. Furthermore, while performance gaps persist between models of different sizes—confirming that parameter scale remains a critical factor in tool-use and reasoning—Search-R2 enables smaller models (e.g., Qwen2.5-7B and Qwen3-8B) to approach the performance of substantially larger models like Qwen2.5-32B-Instruct on tasks such as NQ and TriviaQA. This underscores the framework's efficacy in enhancing search-integrated reasoning for compact models, facilitating their adoption in practical scenarios.

## G. Comparison against Search-R1 with Double Rollout Numbers

To verify that the performance gains of Search-R2 are not merely an artifact of increased rollout volume, we trained the Search-R1 agent with doubled rollouts ($n = 10$, compared to the default $n = 5$). This setting serves as a proxy for a naive refinement strategy where every trajectory is regenerated from scratch, in contrast to Search-R2's targeted refinement of intermediate turns. As shown in Table 7, Search-R2 ($n = 5$, max revision = 1) consistently outperforms Search-R1 ($n = 10$) throughout the training process. At the final step 300, Search-R2 achieves a score of 50.8, surpassing Search-R1 by 6.28%. While increasing $n$ to 10 improves Search-R1, it fails to match the performance of Search-R2. This confirms that our gains stem from the Meta-Refiner's ability to identify and correct specific flaws, rather than simple sample scaling. Furthermore, Search-R2 is significantly more efficient: while Search-R1 ($n = 10$) requires generating 5,120 trajectories per step, Search-R2 generates approximately 3,300 on average, as the Meta-Refiner selects only ∼30% of trajectories for revision. This reduction lowers the computational overhead from 803.2 seconds/step (Search-R1) to 469.5 seconds/step (Search-R2), demonstrating the efficiency of the Meta-Refiner module.

## H. Detailed Ablation Study Results

As a supplement to Section 5.3, we provide the detailed ablation study results on each dataset in Table 8.

### H.1. Trimmer Reliability via Random-Trim Ablation

To verify that the gains of Search-R2 rely on the *learned* trimmer rather than on any extra regeneration, we compare the learned $\pi_h$ against a random-trim baseline that samples the cut-point uniformly from $\{1, \ldots, T\}$ (Table 9). This empirically confirms the "Trimming Skill" condition derived in Section 4: a substantial positive covariance between $\pi_h$ and the regeneration gain $G_k$ is a necessary driver of Search-R2's gains.

### H.2. Scaled Baseline Comparison Across Model Sizes

*Table 8.* The detailed ablation study results of Search-R2 with different base LLMs on seven datasets. The Meta-Refiner, process reward, and joint optimization modules are incorporated into the original Search-R1 framework in an incremental manner.

| Method | NQ | TriviaQA | PopQA | HotpotQA | 2WikiMultiHopQA | Musique | Bamboogle | Average |
|---|---|---|---|---|---|---|---|---|
| **Qwen2.5-7B** | | | | | | | | |
| Search-R1 | 39.5 | 56.0 | 38.8 | 32.6 | 29.7 | 12.5 | 36.0 | 35.0 |
| Search-R1 + Meta-Refiner | 39.3 | 64.4 | 40.3 | 37.0 | 34.6 | 12.7 | 44.0 | 38.9 |
| Search-R1 + Meta-Refiner + Process Reward | 39.6 | 64.9 | 40.5 | 37.4 | 34.9 | 14.2 | 45.6 | 39.6 |
| Search-R2 (Full Version) | 39.9 | 65.9 | 41.0 | 39.0 | 35.8 | 15.1 | 46.2 | 40.4 |
| **Qwen3-8B** | | | | | | | | |
| Search-R1 | 44.0 | 63.1 | 41.8 | 37.2 | 35.5 | 15.7 | 43.0 | 40.0 |
| Search-R1 + Meta-Refiner | 46.2 | 65.9 | 45.1 | 40.2 | 40.2 | 16.2 | 49.6 | 43.4 |
| Search-R1 + Meta-Refiner + Process Reward | 46.7 | 66.4 | 45.6 | 40.7 | 40.6 | 16.7 | 51.0 | 44.0 |
| Search-R2 (Full Version) | 47.7 | 67.6 | 46.6 | 41.2 | 40.5 | 17.2 | 51.2 | 44.6 |
| **Qwen2.5-32B-Instruct** | | | | | | | | |
| Search-R1 | 47.6 | 68.0 | 47.0 | 43.3 | 46.2 | 22.1 | 45.0 | 45.6 |
| Search-R1 + Meta-Refiner | 50.8 | 69.4 | 49.0 | 47.6 | 49.4 | 24.2 | 54.4 | 49.3 |
| Search-R1 + Meta-Refiner + Process Reward | 50.1 | 70.0 | 49.4 | 47.6 | 49.9 | 24.3 | 55.6 | 49.5 |
| Search-R2 (Full Version) | 50.9 | 70.9 | 50.1 | 49.9 | 51.7 | 25.4 | 56.4 | 50.8 |

*Table 9.* Trimmer reliability. The learned trimmer is more than $2\times$ more accurate than random at localizing the earliest flawed step, and replacing it with random trimming degrades end-to-end EM by 3.3–4.1 points consistently across scales.

| Metric / Method | Learned (Ours) | Random Trim | $\Delta$ |
|---|---|---|---|
| Localization Acc. (held-out, %) | **72.1** | 28.3 | +43.8 pp |
| Downstream EM (Qwen2.5-7B) | **40.4** | 36.3 | +4.1 pp |
| Downstream EM (Qwen3-8B) | **44.6** | 41.0 | +3.6 pp |
| Downstream EM (Qwen2.5-32B) | **50.8** | 47.5 | +3.3 pp |

Complementing Table 2 of the main text, in which the non-Search-R1 baselines are reported only on Qwen2.5-7B, we additionally re-ran Rejection Sampling and RAG at the 8B and 32B scales for a strictly apples-to-apples cross-scale comparison (Table 10). The gap between Search-R2 and Search-R1 remains stable at roughly 4–5 EM points across all backbones.

### H.3. Answer-Agnostic Process Reward Ablation

To directly address concerns about hindsight bias in our process-reward judge, we additionally train Search-R2 with an *answer-agnostic* variant of $r_{\text{process}}$, in which the judge is prompted to score

*Table 10.* Average EM across the seven benchmarks, with all baselines extended to the 8B and 32B scales. The relative ordering of methods is preserved at every scale.

| Method | 7B | 8B | 32B |
|---|---|---|---|
| RAG | 30.4 | 34.0 | 39.6 |
| Rejection Sampling | 34.8 | 37.9 | 41.5 |
| Search-R1 | 35.0 | 40.0 | 45.6 |
| **Search-R2 (Ours)** | **40.4** | **44.6** | **50.8** |

query–document relevance from the question and retrieved documents *only*, without access to the ground-truth answer (Table 11). This shows that the dominant source of Search-R2's gain is the Actor-Refiner collaboration and localized correction, not oracle access to the gold answer during reward modeling. We further note that several answer-free reward designs (e.g., evidence coverage/diversity, content-reasoning consistency) are promising future alternatives that we leave to follow-up work.

## I. PPO vs. GRPO Comparison

To further support our choice of GRPO and to confirm that Search-R2's gains are orthogonal to the underlying RL algorithm, we additionally trained Search-R1 and Search-R2 with PPO on two Qwen2.5-7B configurations. Results are summarized in Table 12. Two observations are worth highlighting. First, PPO is unstable in this long-CoT, multi-turn setting: across both configurations we observed a sharp performance drop after roughly 100 training steps, consistent with prior reports on PPO collapse in long-CoT RL training (Yuan et al., 2025). Second, *and most importantly for our contribution*, Search-R2 improves over Search-R1 under both PPO and GRPO (+1.1 to +5.7 EM), confirming that the Actor–Refiner collaboration

*Table 11.* Answer-agnostic vs. answer-aware process reward. The answer-agnostic variant trails the default by only 0.2–0.5 EM points and still substantially outperforms Search-R1 and all retrieval baselines.

| Method | Qwen2.5-7B | Qwen3-8B | Qwen2.5-32B |
|---|---|---|---|
| RAG | 30.4 | 34.0 | 39.6 |
| Rejection Sampling | 34.8 | 37.9 | 41.5 |
| Search-R1 | 35.0 | 40.0 | 45.6 |
| Search-R2 (answer-agnostic $r_{process}$) | **39.9** | **44.2** | **50.6** |
| Search-R2 (answer-aware $r_{process}$, default) | **40.4** | **44.6** | **50.8** |

*Table 12.* PPO vs. GRPO comparison on Qwen2.5-7B. PPO scores reflect peak performance prior to training collapse. Search-R2 improves over Search-R1 under *both* optimizers, confirming that the Actor–Refiner collaboration is orthogonal to the choice of RL algorithm.

| Configuration | GRPO (R1 $\rightarrow$ R2) | PPO (R1 $\rightarrow$ R2) |
|---|---|---|
| Qwen2.5-7B-Base | 35.0 $\rightarrow$ **40.4** (+5.4) | 43.1 $\rightarrow$ **44.2** (+1.1) |
| Qwen2.5-7B-Instruct | 39.6 $\rightarrow$ **45.3** (+5.7) | 38.5 $\rightarrow$ **42.9** (+4.4) |

is orthogonal to the RL optimizer and not an artifact of GRPO. We additionally adopt GRPO in the main experiments because it eliminates PPO's critic network and reduces peak VRAM by approximately 40%, which is essential for fitting the 32B model under our 96 GB GPU budget.

# J. Evaluation on Long-Horizon, Open-Web Benchmarks

To verify that Search-R2 generalizes beyond the standardized E5 + Wikipedia setting of Section 5, we additionally evaluate it on three substantially harder benchmarks: **BrowseComp** and **XBench**, which require long-horizon search-integrated reasoning over noisy, open-web retrieval, and **BrowseComp-Plus**, a fixed but large-corpus deployable variant. Because absolute performance on these benchmarks degrades steeply with model scale, we report results only for the strongest backbone (Qwen2.5-32B) in Table 13.

Search-R2 consistently outperforms Search-R1 with sizable relative gains across all three benchmarks. We emphasize that absolute scores in this regime are inherently low: most open-weight models that do *not* employ specialized agentic tooling (e.g., live web search APIs, code execution sandboxes, file managers, or large-scale web-browsing datasets for training) score below 2.0 on BrowseComp. Frontier systems such as MiroThinker (Team et al., 2025a) and WebSailor (Li et al., 2025a) that *do* employ such tooling can reach 10–20 on BrowseComp, but operate in a fundamentally different evaluation regime than ours. Our setting deliberately fixes a static E5 + Wikipedia retriever and uses only multi-hop QA training data, with no live internet access, web parser, or code execution. The purpose of the comparison here is therefore *out-of-domain generalization of the training method*, not direct competition with end-to-end agentic systems. The strong relative gains under such constrained conditions show that the Actor-Refiner collaboration also transfers to harder, noisier search settings, and that Search-R2 is complementary to system-level agentic engineering: it provides a stronger base policy that such systems can subsequently build upon.

# K. Supplementary Introduction to Trajectory Quality Analysis

## K.1. Rubric Explanation

We evaluate trajectory quality using six rubric dimensions that capture complementary aspects of search-integrated reasoning beyond final answer correctness.

*Evidence Groundedness* measures whether key claims and intermediate conclusions in the trajectory are explicitly supported by retrieved information. A high score indicates that reasoning steps consistently reference or rely on evidence obtained through search, while a low score reflects unsupported claims or hallucinated content.

*Information Density* assesses the usefulness of retrieved information relative to the total search results. Trajectories with high information density primarily retrieve content that directly contributes to solving the task, whereas low scores indicate noisy, weakly relevant, or distracting retrievals.

*Non-Redundancy Efficiency* evaluates how effectively the trajectory uses its search budget. High-scoring trajectories avoid

*Table 13.* Search-R2 vs. Search-R1 on three long-horizon, open-web benchmarks (Qwen2.5-32B backbone). Search-R2 yields large relative improvements across all three, demonstrating out-of-domain generalization beyond the Wikipedia-only training setting.

| Model | BrowseComp | XBench | BrowseComp-Plus |
|---|---|---|---|
| Search-R1 (32B) | 0.5 | 9.3 | 5.1 |
| **Search-R2 (32B)** | **0.9** | **13.4** | **8.3** |
| Relative gain | +80% | +44% | +63% |

*Table 14.* The trajectory quality comparison results among six rubric dimensions on seven datasets. $^\dagger/^\star$ represents in-domain/out-of-domain datasets. All experiments are conducted on the Qwen2.5-32B-Instruct model. In each block of X/Y, X indicates the pair amounts of Search-R2 outperforms Search-R1, while Y indicates the pair amounts of Search-R1 outperforms Search-R2.

| Methods | General QA | | | Multi-Hop QA | | | | |
|---|---|---|---|---|---|---|---|---|
| | NQ$^\dagger$ | TriviaQA$^\star$ | PopQA$^\star$ | HotpotQA$^\dagger$ | 2WikiMultiHopQA$^\star$ | Musique$^\star$ | Bamboogle$^\star$ | Average |
| Evidence Groundedness | 24/1 | 27/0 | 20/1 | 24/4 | 8/6 | 7/2 | 25/3 | 19.3/2.4 |
| Information Density | 37/4 | 28/1 | 35/4 | 40/9 | 37/10 | 32/10 | 46/6 | 36.4/6.3 |
| Non-Redundancy Efficiency | 35/3 | 28/0 | 32/3 | 36/7 | 34/7 | 20/8 | 39/5 | 32.0/4.7 |
| Query Timing Quality | 16/0 | 17/0 | 9/1 | 9/2 | 5/1 | 30/0 | 13/2 | 14.1/0.9 |
| Trajectory Coherence | 35/3 | 29/1 | 32/4 | 34/7 | 30/6 | 23/6 | 36/4 | 31.3/4.4 |
| Uncertainty Handling | 10/0 | 20/1 | 8/1 | 10/2 | 0/5 | 3/2 | 11/1 | 8.9/1.7 |

repeated or unnecessary queries and demonstrate efficient progression toward task-relevant information, while low scores reflect redundant searches or inefficient exploration.

*Query Timing Quality* captures whether searches are issued at appropriate moments and whether the queries are well-formed. High scores correspond to timely searches with precise, informative queries, whereas low scores indicate poorly timed searches or vague and uninformative query formulations.

*Trajectory Coherence* measures the global consistency of the reasoning process. A coherent trajectory maintains alignment between early hypotheses, retrieved evidence, and final conclusions, while incoherent trajectories exhibit logical drift, contradictions, or premature commitment to incorrect assumptions.

*Uncertainty Handling* evaluates how the model responds to incomplete or ambiguous information. High-scoring trajectories appropriately acknowledge uncertainty, seek additional evidence, or hedge conclusions when warranted, whereas low scores indicate overconfident conclusions unsupported by sufficient evidence.

### K.2. Detailed Results

Complementing the analysis in Section 5.6, Table 14 presents the full trajectory quality comparison across seven datasets. Across the six evaluation rubrics, the frequency with which Search-R2 outperforms Search-R1 is significantly higher than the reverse on most datasets. This confirms that our Actor-Refiner collaboration mechanism effectively facilitates the generation of higher-quality search-integrated reasoning trajectories.

### K.3. Evaluation Prompt

To enhance the reproducibility, we provide the prompt for trajectory quality comparison in Table 15.

### K.4. Multi-Evaluator Robustness

To rule out evaluator-specific bias, we re-ran the trajectory comparison protocol described in Section 5.6 using two additional evaluators: (i) `Claude Opus 4.6`, prompted with the same rubric as `GPT-5.1` (Table 15), and (ii) a panel of five human NLP experts, each independently scoring a random subset of 50 pairs per dataset. Table 16 reports aggregated win/fail rates across all seven datasets (700 pairs for the LLM judges, 350 pairs for the human panel).

The win-rate ordering is consistent across evaluators, supporting our claim that the observed trajectory-quality improvements are robust to the choice of judge rather than an artifact of `GPT-5.1` specifically.

## L. Pseudocode for LLM Response Rollout with Multi-Turn Search

We provide the pseudocode for the standard search-integrated reasoning (original Search-R1) in Algorithm 2.

---

**Algorithm 2** LLM Response Rollout with Multi-Turn Search

---

**Require:** Input $x$, policy $\pi_\theta$, search engine $\Lambda$, budget $B$.
**Ensure:** Final response $\hat{y}$.
1: $\hat{y} \leftarrow \varnothing, b \leftarrow 0$
2: **while** $b < B$ **do**
3:     Generate $\hat{y}_b$ until `</search>`, `</answer>`, or EOS.
4:     $\hat{y} \leftarrow \hat{y} + \hat{y}_b$
5:     **if** `<search>` in $\hat{y}_b$ **then**
6:         Extract query $\lambda$; Retrieve $I = \Lambda(\lambda)$
7:         $\hat{y} \leftarrow \hat{y} + $ `<information>`$I$`</information>`
8:     **else if** `<answer>` in $\hat{y}_b$ **then**
9:         **return** $\hat{y}$
10:     **end if**
11:     $b \leftarrow b + 1$
12: **end while**
13: **return** $\hat{y}$

---

## M. Local Process Reward Implementation Details

Local Process Reward ($r_{\text{process}}$) quantifies the *information density* of the retrieved evidence, ensuring that the model is incentivized to perform efficient, non-redundant, and relevant searches. We compute the process reward using a density-based approach evaluated by an external LLM judge (DeepSeek-R1-Distill-Qwen-7B in our experiments). Inference is performed via the vLLM framework with greedy decoding parameters: temperature set to $0.0$, top-p at $0.95$, a repetition penalty of $1.0$, and a maximum token limit of $3,000$. The evaluation procedure proceeds as follows:

1. *Collection Grouping.* For a given reasoning trajectory $y$, we identify all search tool invocations. The top-$k$ documents returned by a single search query are grouped into a single *collection*, denoted as $c_i$. If a trajectory contains $M$ search actions, we have a set of collections $C = \{c_1, \ldots, c_M\}$.

2. *Judge Evaluation.* We construct a prompt provided in Table 17 containing the user question, the ground truth answer, and the chronological list of collections. The judge evaluates each collection $c_i$ against three strict criteria:

   - **Useful ($u_i = 1$):** The collection contains information or clues that help identify the correct answer, even partially.
   - **Not Useful ($u_i = 0$):** The collection is completely irrelevant.
   - **Redundant ($u_i = 0$):** The collection merely duplicates information from previous collections ($c_{1\ldots i-1}$) without adding new insights, even if the information is relevant.

3. *Density Computation.* The judge outputs the total count of useful collections. The process reward is calculated as the ratio of useful collections to total search actions:

$$r_{\text{process}}(y) = \frac{1}{M} \sum_{i=1}^{M} u_i. \tag{16}$$

4. *Outcome Gating.* To prevent "reward hacking" where an agent maximizes retrieval scores without solving the task, the process reward is applied only when the final answer is correct. The total reward $R(y)$ is defined as:

$$R(y) = r_{\text{outcome}}(y) \cdot (1 + r_{\text{process}}(y)), \tag{17}$$

where $r_{\text{outcome}}(y)$ is the binary Exact Match (EM) score.

**Discussion: Hindsight in the Process-reward Judge.** The judge prompt in Table 17 conditions on the ground-truth answer $a_{\text{gold}}$. We use this design because it provides a sharper signal for whether a chunk actually contributes useful

evidence, chunks that are topically related but uninformative for the specific gold answer are filtered out under "Not Useful," while previously-seen information is filtered under "Redundant." We acknowledge that conditioning on $a_{\text{gold}}$ in principle exposes the reward to a form of *hindsight*: the judge benefits from information that the agent did not have at retrieval time. We mitigate and characterize this in three ways: (a) the per-chunk utility is *not* a literal string match against $a_{\text{gold}}$, but a forward-looking judgement on whether the chunk would be useful for inferring the answer; (b) the reward is *gated* by $r_{\text{outcome}}$ (Eq. (1)), so a trajectory cannot be rewarded for retrieving evidence yet producing the wrong answer; and (c) we empirically measure the contribution of the answer-aware signal by ablating it in Appendix H.3, where we find that an answer-agnostic, relevance-based variant of the judge recovers nearly all of Search-R2's improvement ($40.4 \rightarrow 39.9$ on 7B; $50.8 \rightarrow 50.6$ on 32B). This indicates that any residual hindsight bias is a small additive contribution rather than the primary driver of the method's gains. Several answer-free alternatives—e.g., query–document relevance, evidence coverage/diversity, or content–reasoning consistency—are promising future directions for fully eliminating this dependence.

## N. Meta-Refiner Prompt

We provide the prompt for Meta-Refiner in Table 18.

```
SYSTEM: You are a STRICT evaluator for trajectory quality comparison in search-
    integrated reasoning. You will compare two trajectories (A and B) for the SAME
    question.

Rules:
- Do NOT use outside knowledge. Judge only from what the trajectories show.
- Score EACH trajectory independently on each rubric dimension using 0/1/2:
  0=poor, 1=acceptable/mixed, 2=strong.
- If A is slightly better than B on a dimension, assign A a higher score even if
    both are acceptable.
- Avoid giving identical scores unless the two trajectories are truly
    indistinguishable on that dimension.
- After scoring, choose the overall winner (A or B). Output 'Tie' ONLY if nearly
    identical in outcome AND process.
- A/B are just labels.

Return a SINGLE valid JSON object only (no markdown), following the schema exactly.

USER:
QUESTION: {question}

TRAJECTORY A (JSON): {traj_a}

TRAJECTORY B (JSON): {traj_b}

Rubric dimensions: evidence_groundedness, information_density,
    non_redundancy_efficiency, query_timing_quality, trajectory_coherence,
    uncertainty_handling

Output JSON schema (MUST be valid JSON):
{
  "winner": "A" | "B" | "Tie",
  "confidence": 0-100,
  "scores": {
    "A": {
      "evidence_groundedness": 0|1|2,
      "information_density": 0|1|2,
      "non_redundancy_efficiency": 0|1|2,
      "query_timing_quality": 0|1|2,
      "trajectory_coherence": 0|1|2,
      "uncertainty_handling": 0|1|2
    },
    "B": {
      "evidence_groundedness": 0|1|2,
      "information_density": 0|1|2,
      "non_redundancy_efficiency": 0|1|2,
      "query_timing_quality": 0|1|2,
      "trajectory_coherence": 0|1|2,
      "uncertainty_handling": 0|1|2
    }
  },
  "reasons": [ "reason1", "reason2", "reason3" ]
}

Decision procedure:
1) Score A and B independently on all rubric dimensions.
2) Decide winner primarily by outcome quality, if similar, decide by process quality
    . 3) Use 'Tie' only if truly indistinguishable.

Notes: - reasons: at most 3 short bullet strings.
```

*Table 15.* Prompt for trajectory quality comparison.

*Table 16.* Trajectory-quality preferences across three evaluators. Across every evaluator, Search-R2 is preferred over Search-R1 by a 5–7× margin. Inter-annotator agreement is high under Fleiss' $\kappa$: 0.68 among the five human experts and 0.71 across the two LLM judges.

| Evaluator | Win (%) | Fail (%) | Tie (%) |
|---|---|---|---|
| GPT-5.1 | 23.7 | 3.4 | 72.9 |
| Claude Opus 4.6 | 28.7 | 4.1 | 67.2 |
| Human Experts ( $n$=5 ) | 21.4 | 4.5 | 74.1 |

```
SYSTEM:
You are evaluating numbered collections of retrieved documents to determine their
    usefulness in answering a given question, where each collection is enclosed
    within <collection_x> and </collection_x> tags containing documents that belong
    to that collection. Given the question and its correct answer, mark each
    collection as "useful" (yes) or "not useful" (no) based on these criteria:
(1) A collection is useful if it contains information or clues that help identify
    the correct answer, even partially.
(2) A collection is not useful if it's completely irrelevant to the question.
(3) A collection is not useful if it merely duplicates information from previous
    collections without adding new insights, even if that information would
    otherwise be relevant. After evaluating all collections strictly according to
    these criteria and the provided information, report the total count of
    collections marked as useful.

USER:
Question: {question}

Answer: {answer}

{M} collections of the tool responses:

<collection_1>
Doc 1: {retrieved_info_1_top_1}
Doc 2: {retrieved_info_1_top_2}
......
Doc k: {retrieved_info_1_top_k}
</collection_1>

......

<collection_{M}>
......
Doc k: {retrieved_info_{M}_top_k}
</collection_{M}>

Provide your answer strictly in the format:
Final Answer: number
```

*Table 17.* LLM Judge Prompt for Chunk Utility $u_i$. The system prompt enforces strict criteria for relevance and non-redundancy.

```
You are a meticulous meta-thinker. Review the numbered ASSISTANT_STEP entries and
    identify the earliest flawed step. Return a single integer between 0 and {
    max_steps} where 0 means all steps are acceptable.

ASSISTANT_CONTEXT: <assistant turns before current rollout>

USER: <user message>

ASSISTANT_STEP_1: <assistant turn 1>
TOOL: <tool output triggered by step 1 (if any)>

ASSISTANT_STEP_2: <assistant turn 2>
TOOL: <tool output triggered by step 2 (if any)>

...

Problematic step index (0 = no issue):
```

*Table 18.* Prompt for Meta-Refiner.

