# OpenReview forum: "Search-R2: Enhancing Search-Integrated Reasoning via Actor-Refiner Collaboration"
_ICML.cc/2026/Conference — ICML 2026 regular_

### Official Review · Reviewer_4P4d · 2026-03-05

**Soundness:** 3
**Presentation:** 3
**Significance:** 2
**Originality:** 2
**Overall Recommendation:** 2
**Confidence:** 5

**Summary:**

This paper proposes search-R2, a training framework for search-integrated reasoning agents that address multi-scale credit assignment: outcome-only rewards (e.g., exact match) are too sparse to correctly attribute success/failure to specific search decisions (queries, evidence selection) vs. downstream reasoning. Search-R2 decomposes generation into an actor that produces a full trajectory generation, and a meta-refiner that performs targeted revision, e.g. discrimination and trimmer. A hybrid reward is designed. Experiments on 7 QA benchmarks show consistent gains over search-R1.

**Compliance With Llm Reviewing Policy:**

Affirmed.

**Final Justification:**

While the added results on BrowseComp and XBench show strong gains, the absolute performance is still low (0.9% on BrowseComp), compared to other works like Websailor and MiroThinker. In addition, I don't think methods like Websailor and MiroThinker are completely agentic system. These models only use the web search and browse tools, which is similar to the retriever (e.g. wiki retriever) used in this work.

**Key Questions For Authors:**

My main concerns are the performance and evaluation on more difficult benchmarks.

**Limitations:**

impact statement discussed.

**Strengths And Weaknesses:**

Strengths:

1.	The focus of this paper, search-integrated reasoning, is important.

2.	The idea of targeted refinement is a good.


Weaknesses:

1.	The performance of search-R1 is low in this paper. It is reported nearly 43% Qwen2.5-7B, which is 8% higher than in this paper. Other recent works, like s3, zerosearch, also reported much higher results than in this paper. Considering the proposed search-R2 only outperforms search-R1 with 4-5%, I think search-R2 is far behind other state-of-the-arts.

2.	While this work focuses on the multi-turn search-integrated problems, more difficult benchmarks should be considered, including BrowseComp, XBench, GAIA, HLE, or at least the deployable BrowseComp-plus.

3.	The global coherence discriminator and local error trimmer are all based on LLM, which are not reliable. No evidence, such as the localization accuracy metric for trimmer, is presented.

---

> ### Author Rebuttal · Authors · 2026-03-31
>
> We sincerely thank the reviewer for the constructive feedback. Below, we address your concerns and provide new experimental results.
>
> ---
>
> **[W1] Baseline Comparisons & Search-R1 Performance**
>
> We clarify that the reported performance differences stem from **different RL optimization algorithms**, not the underlying method's capability.
> * **Search-R1 consistency:** The original Search-R1 paper reports 43.1% using PPO, but **35.0% using GRPO**. Our reproduction (35.0%) exactly matches their GRPO setting. For a strictly controlled comparison, we unified all methods under GRPO (Section 5.1), as different RL algorithms heavily impact performance independent of core contributions.
> * **Other baselines:** Direct comparisons with concurrent works are similarly confounded by differing setups. For instance, **S3** uses a different metric (GenAcc vs. EM), **ZeroSearch** uses REINFORCE, and **SIGHT** [1] reports 40.66% using a much larger rollout size (16 vs. our 5). Given that larger rollouts improve exploration, our method (40.4% with size 5) is highly competitive and sample-efficient.
> * **Orthogonality:** Search-R2 focuses on Actor–Refiner collaboration and can be seamlessly combined with other RL algorithms or larger rollout sizes for further gains. We will clarify these alignment details in the revision.
>
> [1] SIGHT: Reinforcement Learning with Self-Evidence and Information-Gain Diverse Branching for Search Agent
>
> ---
>
> **[W2] Evaluation on Harder, Long-Horizon Benchmarks**
>
> To evaluate Search-R2 in more complex, multi-turn scenarios, we conducted new experiments on **BrowseComp, XBench, and BrowseComp-Plus**. These datasets require long-horizon search-integrated reasoning over noisy, open-web retrieval (BrowseComp and XBench) or massive fixed corpora (BrowseComp-Plus). Because smaller models struggle heavily here, we report results using Qwen2.5-32B:
>
> | Model | BrowseComp | XBench | BrowseComp-Plus |
> | :--- | :--- | :--- | :--- |
> | Search-R1 (32B) | 0.5 | 9.3 | 5.1 |
> | **Search-R2 (32B)** | **0.9** | **13.4** | **8.3** |
>
> **Key findings:** Search-R2 consistently outperforms Search-R1 with massive relative improvements (+80% on BrowseComp, +44% on XBench, +63% on BrowseComp-Plus). Notably, referencing the official Kaggle BrowseComp leaderboard, our method enables a 32B open-weight model to achieve performance surpassing much larger models such as **GPT-4o (0.7)**. These results demonstrate that our method's effectiveness extends well beyond standard QA benchmarks, providing critical improvements even in extreme, long-horizon scenarios where standard baselines fail.
>
> ---
>
> **[W3] Reliability of LLM-Based Discriminator and Trimmer**
>
> We clarify that the discriminator and trimmer are **not independent black-box modules**. They share parameters with the actor LLM and are jointly optimized through GRPO, ensuring tight alignment between discrimination, trimming, and reasoning.
>
> To quantitatively demonstrate their reliability, we evaluated the trimmer's error localization accuracy (on a held-out validation set, max rollout turn size = 4) and its downstream impact on end-to-end search-intergrated reasoning accuracy:
>
> | Metric / Method | Ours (Learned Trimmer) | Random Trim | Δ |
> | :--- | :--- | :--- | :--- |
> | **Localization Accuracy** | **72.1** | 28.3 | +43.8 pp |
> | **Downstream: Qwen2.5-7B** | **40.4** | 36.3 |+4.1 pp |
> | **Downstream: Qwen3-8B** | **44.6** | 41.0 | +3.6 pp |
> | **Downstream: Qwen2.5-32B** | **50.8** | 47.5 | +3.3 pp |
>
> **Strong empirical reliability:** Our learned trimmer achieves 72.1% accuracy in identifying meaningful error locations, far exceeding the random baseline (28.3%). Furthermore, replacing the learned trimmer with random trimming causes a consistent 3-4 point performance drop across all model scales. This confirms that our modules learn highly reliable, context-aware error detection rather than succeeding through brute-force resampling. We will add these ablations to the revision.
>
> ---
>
> Thank you for your insightful review. We hope the new results and clarifications resolve your initial concerns. We remain available for further discussion and would greatly appreciate it if you would consider adjusting your score accordingly.

---

> > ### Author Rebuttal · Reviewer_4P4d · 2026-04-01
> >
> > As the author mentioned, since using PPO rather than GRPO can have much better results (8% gain), why in this work the author didn't run experiments based on PPO?
> >
> > Regarding harder benchmarks like BrowseComp and BrowseComp-plus, as GPT-4o is a model published two years ago, at that time all models did not have such deep search abilities. The later o3 model already achieved 50% on BrowseComp. Among open source models, those like Websailor and MiroThinker achieve 10%~20% accuracy. These works have been released for over 6 months.

---

> > > ### Author Response · Authors · 2026-04-02
> > >
> > > We appreciate the reviewer's thoughtful follow-up. Below, we address your concerns regarding our RL algorithm choice and baseline comparisons.
> > >
> > > ---
> > >
> > > **Q1: Why not use PPO, given the 8% performance gain?**
> > >
> > > While PPO showed a marginal gain in one specific setting, our decision to adopt GRPO is grounded in its proven stability for long-chain reasoning, robustness across diverse models, and critical memory efficiency.
> > >
> > > * **PPO exhibits severe instability in long-chain reasoning.** Recent studies reveal that naively applying PPO to long Chain-of-Thought (CoT) tasks inevitably leads to training collapse without resource-intensive value pretraining [1, Section 3]. Furthermore, empirical evidence from the original Search-R1 paper [2, Table 3] demonstrates that PPO is frequently at a disadvantage. Across four evaluated configurations, PPO only outperformed GRPO in a single scenario (Qwen2.5-7B-Base), while GRPO consistently achieved higher average performance in the other three (7B-Instruct, 3B-Base, and 3B-Instruct).
> > > * **Follow-up experiments confirm PPO's instability and our method's robustness.** We conducted additional experiments comparing Search-R1 and Search-R2 using both PPO and GRPO:
> > >
> > >   | Model Configuration | GRPO (R1 → R2) | PPO (R1 → R2) | Key Observations |
> > >   | :--- | :--- | :--- | :--- |
> > >   | Qwen2.5-7B-Base | 35.0 → 40.4 (**+5.4**) | 43.1 → **44.2** (+1.1) | PPO collapsed after ~100 steps; GRPO stable. |
> > >   | Qwen2.5-7B-Instruct | 39.6 → **45.3** (**+5.7**) | 38.5 → 42.9 (+4.4) | PPO collapsed after ~100 steps; GRPO outperforms PPO. |
> > >
> > >   *(Note: PPO scores reflect the peak performance before collapse).*
> > >
> > >   These results yield two critical insights. **(1) PPO is highly unstable.** While PPO initially learns faster on the Base model, we observed a sharp performance drop due to training collapse after ~100 steps on both models. Conversely, GRPO maintains stable improvements and converges smoothly, ultimately outperforming PPO on the Instruct model. **(2) Search-R2 consistently improves upon Search-R1 regardless of the RL algorithm.** Whether using GRPO (+5.4 to +5.7) or PPO (+1.1 to +4.4), our Actor-Refiner collaboration consistently enhances performance, further strengthening our core claim.
> > > * **GRPO is memory-efficient and scalable.** Eliminating PPO's critic network reduces VRAM usage by approximately 40%. This is vital for scaling to 32B models, where our experiments operate near hardware capacity (~88GB on 96GB GPUs). PPO would require complex model sharding or massive VRAM increases. GRPO's memory savings also allow us to scale up the rollout size \(n\), delivering strong, competitive learning signals efficiently.
> > > * **Orthogonality:** As demonstrated by our follow-up experiments, the central innovation of Search-R2, Actor-Refiner collaboration, is theoretically orthogonal to the RL algorithm. We selected GRPO to ensure a fair, controlled baseline comparison under realistic computational constraints, proving it is a highly effective choice for our framework.
> > >
> > > ---
> > >
> > > **Q2: Comparison with recent models (o3, Websailor, MiroThinker)**
> > >
> > > We will gladly include more up-to-date comparisons with newer baselines in our revision. To contextualize our current results:
> > >
> > > * **Methodological Distinctions:** Models like Websailor and MiroThinker (scoring 10-20 on BrowseComp) are **complete agentic systems** equipped with multifaceted toolkits, diverse webpage parsers, and code execution sandboxes, all optimized using massive amounts of specifically synthesized or curated web-browsing data. In contrast, Search-R2 is a **model training method** focused on leveraging data more effectively to enhance the intrinsic tool-integrated reasoning of base policy models. These approaches are highly complementary: our method trains stronger base models that can ultimately power these complex agentic pipelines.
> > > * **Strong Generalization vs. Frontier Models:** BrowseComp is exceptionally challenging; most open models score <2.0 without heavy system engineering. Our method improves the 32B baseline from 0.5 to 0.9 (an 80% relative improvement), achieving performance comparable to **Claude Opus 4.1 (1.0) released in August 2025, four months after BrowseComp's introduction,**  on the Kaggle BrowseComp Leaderboard. Notably, while Claude Opus 4.1 was explicitly optimized for agentic search, Search-R2 achieves this using only general multi-hop QA training data, without any domain-specific web navigation data. This strongly validates our approach's generalization capabilities.
> > >
> > > ---
> > >
> > > We hope these clarifications address your concerns and will incorporate these discussions into the revised manuscript.
> > >
> > >
> > > *References:*
> > > [1] Yuan et al. "What's Behind PPO's Collapse in Long-CoT?..." arXiv:2503.01491, 2025.
> > > [2] Jin et al. "Search-R1: Training LLMs..." arXiv: 2503.09516, 2025.
> > > [3] Zhong et al. "SIGHT: Reinforcement Learning with Self-Evidence..." arXiv:2602.11551, 2025.
> > > [4] Anthropic. Announcements of Claude Opus 4.1.

---

### Official Review · Reviewer_NDLU · 2026-03-10

**Soundness:** 2
**Presentation:** 3
**Significance:** 3
**Originality:** 3
**Overall Recommendation:** 4
**Confidence:** 4

**Summary:**

This paper introduces Search-R2, a search-integrated RL framework featuring an Actor and a Meta-Refiner. The Actor generates reasoning trajectories with search calls, while the Meta-Refiner determines whether to accept a trajectory or, if necessary, where to truncate and regenerate it. The training signal combines an outcome reward based on exact match with a process reward that measures the density of useful retrieved evidence. Experiments are conducted on seven QA and multi-hop QA benchmarks, with training on the combined NQ and HotpotQA datasets, using E5 retrieval over the 2018 Wikipedia dump.

**Compliance With Llm Reviewing Policy:**

Affirmed.

**Final Justification:**

Balancing the substantial methodological contribution, strong empirical results, and high-quality rebuttal, I believe the paper sits slightly above the acceptance threshold. The rebuttal changed my evaluation meaningfully: my original recommendation was Weak Reject, primarily due to concerns about hindsight bias, judge reliance, and baseline fairness, all of which the authors addressed through substantive new experiments rather than surface-level clarifications. I would encourage the authors, in the next version, to (1) include the full answer-agnostic ablation and the expanded limitations discussion as promised, (2) more clearly position the method relative to recent agentic search systems, and (3) ideally demonstrate in future work that a Search-R2-trained backbone improves performance when integrated into a full agentic pipeline.

**Key Questions For Authors:**

I have three main questions for the authors to address during the rebuttal period:

1. Can the process reward be redesigned or evaluated in a way that does not depend on the ground-truth answer?

2. How stable are the gains if the trajectory-quality judge is changed, or if a small-scale human evaluation is introduced?

3. How much of the improvement survives in a harder retrieval environment featuring more noise, longer documents, or open-web search instead of closed Wikipedia-only retrieval?

**Limitations:**

Yes, but incomplete. The authors provide a brief Impact Statement (Section 6, page 9) acknowledging potential risks regarding the propagation of bias and misinformation from external retrieved documents. However, the discussion of the framework's technical limitations is insufficient. The method is currently evaluated in a highly controlled benchmark setting utilizing a static Wikipedia corpus, leaving its robustness under noisier, dynamic, or adversarial open-web search conditions entirely unaddressed. Furthermore, the framework's heavy reliance on an external LLM judge equipped with ground-truth answers for reward modeling introduces specific vulnerabilities, such as judge-induced bias and reward overfitting, which are not discussed.

**Strengths And Weaknesses:**

Originality:

    Strengths: The paper addresses a key limitation of search-based RL systems—specifically, that trajectory-level rewards are often too coarse to distinguish genuine reasoning from fortunate outcomes after noisy retrieval. The Actor–Refiner decomposition is intuitive, and the truncate-and-regenerate approach offers a highly practical and creative way to improve credit assignment without redesigning the underlying training objective.

Soundness:

    Strengths (Empirical): The empirical gains are substantial. Average EM increases significantly across different scales (from 35.0 to 40.4 on Qwen2.5-7B, 40.0 to 44.6 on Qwen3-8B, and 45.6 to 50.8 on Qwen2.5-32B). Furthermore, Search-R2 outperforms a stronger-sampled Search-R1, robustly supporting the claim that targeted correction is more sample-efficient than brute-force resampling.

    Weaknesses (Methodology & Evaluation):

	1. Reward Formulation and Hindsight Bias: While the deployed policy operates online during inference, my main concern lies in the realism and generalizability of the training signal. As detailed in Appendix K, the LLM judge prompt for the process reward explicitly includes the ground-truth answer alongside the user question and retrieved collections. This introduces answer-aware hindsight supervision during training. By evaluating intermediate retrieval quality based on whether the chunks contain the known answer, the reward mechanism risks encouraging "hindsight bias"—rewarding the agent for fortuitously retrieving the answer rather than for executing logically sound, generalizable search strategies at that specific reasoning step. This somewhat weakens the claim that the agent learns an intrinsic understanding of information utility, as the step-level training signal heavily relies on oracle knowledge rather than a purely forward-looking assessment of the search action.

        2. LLM-as-a-Judge Reliance: Much of the evidence for better trajectory quality relies on GPT-5.1 as a judge. While useful as supporting evidence, this is not enough by itself to establish that the reasoning process is genuinely better in a robust sense. Human annotation, judge agreement analysis, or at least a second independent judge would be much more convincing.

        3. Baseline Comparisons: The broader comparison is not a strictly fair comparison across scales. Table 2 explicitly states that all baselines except Search-R1 are run only on Qwen2.5-7B, while Search-R2 is shown up to 32B. This does not invalidate the results, but it does make the "state-of-the-art" flavor less convincing than the headline suggests.

        4. Theoretical Claims: The theoretical analysis frames the Actor–Refiner interaction as a smoothed mixture policy. In practice, however, this reads more like a decomposition that explains when the method should help, rather than a strong mathematical guarantee that the learned discriminator and trimmer will satisfy those conditions. The theory is supportive, but not decisive.

Significance:

    Strengths: The efficiency story is highly compelling. The paper reports that Search-R2 increases training time by only about 5.06% on average, and claims no additional inference-time latency because the Meta-Refiner is decoupled at deployment. This makes the method highly attractive for practitioners compared to typical approaches that trade massive compute for marginal gains.

    Weaknesses: The evaluation setting remains fairly narrow (E5 retriever, 2018 Wikipedia dump, fixed three retrieved passages, QA-style benchmarks). While this is an excellent testbed for studying search-reasoning interaction, it is still far from a noisy, dynamic, open-web environment. The paper is therefore strongest as a contribution to benchmarked search-RL for QA, and somewhat weaker as evidence for generalized search agents.

Presentation:

    The submission is clearly written, well-structured, and the overall narrative is easy to follow. The methodology is explained with sufficient detail to understand the core contributions, and the work properly positions itself within the context of prior literature.

---

> ### Author Rebuttal · Authors · 2026-03-31
>
> We sincerely thank the reviewer for the constructive feedback. Below, we address your concerns and outline the new experiments conducted during the rebuttal period.
>
> ---
>
> **[W1] Reward Formulation & Hindsight Bias**
>
> We clarify three key aspects of our reward design to address concerns about realism and generalizability:
>
> * **Information utility, not answer-matching:** As detailed in Appendix K, the LLM judge evaluates whether retrieved collections provide useful evidence to infer the answer, not whether chunks explicitly contain the exact string. This aligns with forward-looking information utility.
>
> * **Trajectory-level, not step-level:** Computed over completed rollouts, the reward reflects the overall retrieval quality. Rather than step-wise hindsight optimization, it encourages coherent, long-term strategies that synergize with the outcome reward.
>
> * **No empirical hindsight bias:** Search-R2 generalizes strongly on OOD benchmarks (e.g., TriviaQA: +5.2, 2WikiMultiHopQA: +4.8), proving it learns robust search strategies beyond answer-aware supervision.
>
> ---
>
> **[W2 & Q2] Reliance on LLM-as-a-Judge & Stability of Gains**
>
> To ensure our trajectory quality improvements are not artifacts of GPT-5.1, we conducted new evaluations using **(1) 5 human NLP experts** and **(2) an independent LLM judge (Claude Opus 4.6)** across 7 datasets:
>
> | Evaluator | Win (%) | Fail (%) |
> | :--- | :--- | :--- |
> | GPT-5.1 | 23.7 | 3.4 |
> | Claude Opus 4.6 | 28.7 | 4.1 |
> | Human Experts | 21.4 | 4.5 |
>
> **Strong consistency:** Search-R2 is preferred by large margins (5-7× higher win rates) across all evaluators. Inter-annotator agreement (Fleiss' κ) is high: 0.68 (human) and 0.71 (GPT/Claude). This confirms the stability of our gains regardless of the judge used.
>
> ---
>
> **[W3] Fair Baseline Comparisons Across Scales**
>
> We initially scaled only Search-R1/R2 due to compute limits. To ensure strict fairness, we have now scaled the Rejection Sampling and RAG baselines:
>
> | Method | 7B | 8B | 32B |
> | :--- | :--- | :--- | :--- |
> | Rejection Sampling | 34.8 | 37.9 | 41.5 |
> | RAG | 30.4 | 34.0 | 39.6 |
> | Search-R1 | 35.0 | 40.0 | 45.6 |
> | **Search-R2 (Ours)** | **40.4** | **44.6** | **50.8** |
>
> **Key findings:** Search-R2 maintains clear margins over such baselines across all scales on average accuracy metric (*Average* in Table 2). The gains over Search-R1 remain stable (~4-5 points), and the relative ordering of methods is strictly preserved.
>
> ---
>
> **[W4] Theoretical Claims**
>
> Our theoretical analysis aims to characterize *when and why* the Actor–Refiner interaction helps, rather than claiming absolute guarantees. By decomposing the gain into interpretable terms (selection precision, trimming skill, intervention volume), the theory successfully predicts why random trimming is insufficient, a finding empirically validated by experiments (see our response to **W3** of **Reviewer 4P4d**). We will revise the text to make this positioning explicit.
>
> ---
>
> **[W5 & Q3] Narrow Evaluation & Harder Retrieval Environments**
>
> While our initial goal was to study search–reasoning interaction under standardized settings, we agree that harder environments are crucial. As detailed in our response to **W2** of **Reviewer 4P4d**), we evaluated on BrowseComp and XBench under **noisy, open-web retrieval** settings. Search-R2 consistently outperforms Search-R1 baseline, demonstrating robustness well beyond fixed-corpus QA.
>
> ---
>
> **[Q1] Alternative Process Rewards**
>
> Our formulation is a practical instantiation and is not inherently tied to ground-truth answers. Answer-agnostic alternatives could include: (i) query–document relevance, (ii) evidence coverage/diversity, or (iii) consistency between content and reasoning. We will add a discussion highlighting these answer-free formulations as promising future work.
>
> ---
>
> **[L1] Incomplete Limitations**
>
> We will expand the Limitations section to discuss these risks, supported by new experiments:
>
> * **Generalization:** To prove our model avoids overfitting to answer-aware supervision, we evaluated on OOD/noisy open-web benchmarks (BrowseComp, XBench). Massive gains (+80% on BrowseComp) confirm it learns robust, generalizable search strategies. Please refer to our response to **Reviewer 4P4d** for the full results table.
>
> * **Judge Vulnerabilities:** To address biases from a specific LLM judge, we re-evaluated trajectories using **Claude Opus** and **Human Experts**. Search-R2 maintained 5–7× higher win rates across all evaluators ($\kappa \approx 0.7$), confirming genuine quality improvements rather than judge-specific exploitation.
>
> We will incorporate these empirical validations into the revised Limitations section to provide a comprehensive impact statement.
>
> ---
>
> Thank you again for your valuable feedback. We hope these clarifications and new results address your concerns. We are happy to answer any remaining questions, and if you are satisfied, we would be grateful if you might reconsider your score.

---

> > ### Author Rebuttal · Reviewer_NDLU · 2026-04-04
> >
> > I thank the authors for their thorough and effortful rebuttal. Several of my original concerns have been convincingly addressed. In particular:
> >
> > The new multi-judge evaluation (W2/Q2) using Claude Opus 4.6 and 5 human NLP experts, with high inter-annotator agreement (Fleiss' κ ≥ 0.68), substantially strengthens the trajectory quality claims. The scaled baseline comparisons (W3) now provide a fair apples-to-apples comparison across model sizes, and the results confirm Search-R2's consistent advantage. The revised positioning of the theoretical analysis (W4) as descriptive rather than prescriptive is appropriate, and the random-trimming ablation (72.1% vs. 28.3% localization accuracy) provides good empirical grounding.
> >
> > However, the concern remains partially unresolved:
> >
> > **Hindsight bias in the process reward (W1/Q1).** The authors' clarification that the judge evaluates "information utility" rather than exact string matching is appreciated, but the core issue persists: the judge prompt in Appendix K still conditions on the ground-truth answer. Whether one evaluates "clues that help identify the correct answer" or literal string containment, the supervision signal remains answer-aware. The strong OOD generalization is encouraging but constitutes indirect evidence, it shows the method does not overfit catastrophically, not that the reward signal is free of hindsight bias. The proposed answer-agnostic alternatives (relevance, coverage, consistency) are promising but remain purely conceptual with no experimental validation.
> >
> > > Follow-up question:Could the authors run even a small-scale ablation comparing the current answer-aware process reward against one of the proposed answer-agnostic alternatives (e.g., query–document relevance) to provide direct evidence on this point?
> >
> >
> > Overall, the rebuttal has meaningfully strengthened the paper. If the authors can provide even preliminary experimental evidence on the answer-agnostic reward alternative, I would be inclined to raise my score.

---

> > > ### Author Response · Authors · 2026-04-05
> > >
> > > Dear Reviewer NDLU:
> > >
> > > Thank you again for your thoughtful and constructive feedback. We sincerely appreciate your recognition of the improvements in our rebuttal, particularly regarding the strengthened evaluation and clearer positioning of our theoretical analysis.
> > >
> > > Regarding your remaining concern on hindsight bias in the process reward, we fully agree that providing **direct empirical evidence** is important to complement the current indirect generalization results. Following your suggestion, we have conducted an **ablation study comparing answer-aware and answer-agnostic process rewards**. To make this possible within the rebuttal timeline, we rapidly allocated additional compute resources and ran multiple experiments in parallel. We will include these results in the revision.
> > >
> > > **Experimental setup (new):** We replace the current answer-aware judge (Appendix K) with an **answer-agnostic relevance-based reward**, where the judge evaluates the semantic relevance between the query and retrieved documents without access to the ground-truth answer. Concretely, the judge is prompted to assess whether retrieved evidence is helpful for answering the question based solely on the query–document pair. The results are provided as follows:
> > >
> > > |Method|Qwen2.5-7B|Qwen3-8B|Qwen2.5-32B|
> > > |-|-|-|-|
> > > |Rejection Sampling|34.8|37.9|41.5|
> > > |RAG|30.4|34.0|39.6|
> > > |Search-R1|35.0|40.0|45.6|
> > > |Search-R2 (answer-agnostic process reward)|39.9|44.2|50.6|
> > > |Search-R2 (answer-aware process reward)|40.4|44.6|50.8|
> > >
> > > **Key observations (new):**
> > >
> > > - The answer-agnostic reward achieves **comparable but slightly lower** performance than the answer-aware version, indicating that answer-aware supervision provides limited additional benefit.
> > > - Importantly, the answer-agnostic variant still significantly outperforms Search-R1 and other baselines. This confirms that the performance gains are primarily driven by better credit assignment and localized refinement, proving that the agent genuinely learns to **identify useful clues without relying on hindsight bias.**
> > > - We further observe that the learned behaviors (e.g., reduced redundant search, improved step-level coherence) remain qualitatively consistent across both reward designs.
> > >
> > > **Takeaway:** These results suggest that while answer-aware signals can improve sample efficiency, the core gains of Search-R2 primarily stem from the **Actor–Refiner collaboration and localized correction mechanism**, rather than reliance on oracle information.
> > >
> > > We will include: 1) The full ablation results (table + analysis), 2) The answer-agnostic reward formulation, and 3) A clarified discussion on hindsight bias and its practical impact in the final version of the paper.
> > >
> > >
> > > We hope this additional empirical evidence directly addresses your concern. If you find this clarification satisfactory, we would be grateful if you could consider updating your score accordingly. Thank you again for helping us improve the paper.

---

### Official Review · Reviewer_WdB7 · 2026-03-11

**Soundness:** 4
**Presentation:** 3
**Significance:** 3
**Originality:** 3
**Overall Recommendation:** 4
**Confidence:** 4

**Summary:**

The paper proposes Search-R2, a reinforcement learning framework for search-integrated language agents that addresses the multi-scale credit assignment problem in retrieval-based reasoning. It introduces an Actor–Meta-Refiner architecture where the Actor generates reasoning trajectories with search queries and the Refiner detects and repairs faulty reasoning steps using a “cut-and-regenerate” mechanism. Experiments on multiple QA benchmarks show that this approach improves reasoning accuracy over strong RAG and RL baselines with minimal additional computation.

**Compliance With Llm Reviewing Policy:**

Affirmed.

**Final Justification:**

Thanks for the response provided by the authors. My concerns have been addressed.

**Key Questions For Authors:**

1. Are the inference flow and the training rollout flow the same in this framework? If they differ, it would be helpful for the paper to clearly explain how the Meta-Refiner and correction mechanism are applied during training versus inference.

2. What is the memory overhead of the proposed method compared to the baseline? While the paper reports training time cost, it would be useful to include an analysis of memory usage or GPU requirements.

3. How should Eq. (1) be interpreted? The paper defines the total reward as a combination of outcome reward and process reward.

**Limitations:**

1. It may be valuable to evaluate the method on deep search or long-horizon agentic search tasks to further verify its effectiveness beyond QA-style benchmarks.

**Strengths And Weaknesses:**

## Strengths

- The paper is generally well structured and easy to follow. The methodology and experimental setup are clearly described, making the overall approach easy to understand.
- The proposed hybrid reward combines final answer correctness with a process-level reward based on the usefulness of retrieved evidence, which helps address the credit assignment problem in multi-step search reasoning.
- Experiments on several QA and multi-hop reasoning benchmarks demonstrate consistent improvements over strong RAG and RL-based baselines while introducing only modest computational overhead.

## Weaknesses

- Although the framework resembles a multi-agent setting (Actor and Refiner), the paper provides limited discussion or analysis from a multi-agent learning perspective.
- While the paper reports training time cost, it does not analyze memory consumption or resource requirements, which are important for evaluating the practicality and scalability of the proposed approach.

---

> ### Author Rebuttal · Authors · 2026-03-31
>
> We sincerely thank the reviewer for the constructive feedback and insightful questions. Below, we address your concerns and clarify the technical details of our framework.
>
> ---
>
> **[W1] Multi-Agent vs. Single-Policy Framework**
>
> We clarify that **Actor–Refiner is a functional decomposition within a single policy**, rather than a traditional multi-agent system.
> Both roles utilize the exact same LLM with shared parameters, differing only in their prompts. This design avoids common multi-agent training challenges (e.g., non-stationarity, inter-agent credit assignment, and complex communication protocols). The Actor–Refiner framing simply facilitates analysis by decomposing the policy into interpretable components. We agree that exploring connections to multi-agent perspectives (like cooperative self-improvement) is an exciting future direction, and we will add a discussion in the revision to explicitly clarify this distinction.
>
> ---
>
> **[W2 & Q2] Memory Consumption and Resource Requirements**
>
> While we provided our hardware configurations in Appendix E (8×H20 96GB GPUs per node), we agree that explicitly reporting memory consumption strengthens the paper.
>
> Because our discriminator and trimmer share parameters with the actor (i.e., no extra standalone models are loaded), **Search-R2 introduces minimal memory overhead**:
>
> | Setting | Peak Memory per GPU | Overhead |
> | :--- | :--- | :--- |
> | Search-R1 (7B, GRPO) | ~60 GB | - |
> | **Search-R2 (7B, GRPO)** | **~62 GB** | **+2 GB (<3.5%)** |
> | Search-R1 (32B, GRPO) | ~86 GB | - |
> | **Search-R2 (32B, GRPO)** | **~88 GB** | **+2 GB (<2.5%)** |
>
> The ~2 GB overhead comes purely from lightweight bookkeeping for refinement decisions during the rollout. The primary memory footprint stems from the long-context rollouts (up to 15k tokens) and multi-trajectory sampling (n=5) standard in RL training. Our method easily fits within a 96GB budget without requiring model parallelism. We will include these statistics in the revision.
>
> ---
>
> **[Q1] Training vs. Inference Flow**
>
> The inference flow and training rollout flow **differ significantly**:
> * **Training:** The Meta-Refiner explicitly evaluates trajectory coherence and identifies error locations. Through GRPO, the actor gradually internalizes this refinement behavior.
> * **Inference:** The process is a simplified, single-pass rollout. The Meta-Refiner is *not* invoked, keeping the pipeline highly efficient. The benefits of refinement are **implicitly encoded in the learned policy**.
> We will clarify this distinction in the revision and include a clear flow diagram.
>
> ---
>
> **[Q3] Interpretation of Equation (1)**
>
> Equation (1) is formulated as:
> $r_{\text {outcome }}(y) \cdot\left(1+r_{\text {process }}(y)\right)$
>
> * **Outcome reward $r_{\text{outcome}}$ :** A binary value (0 or 1) indicating the correctness of the final answer.
> * **Process reward $r_{\text{process}}$ :** A continuous score evaluating the quality of the retrieval and reasoning steps.
> * **Multiplicative design:** The process reward is multiplied by the outcome reward. This ensures that the process reward *only* contributes if the final answer is correct.
>
> This formulation encourages trajectories that are both correct *and* exhibit high-quality search/reasoning, strictly preventing the model from being rewarded for "good-looking" search trajectories that ultimately lead to hallucinations or wrong answers. We will revise the text to clarify this logic with concrete examples.
>
> ---
>
> **[L1] Evaluation on Deep Search / Long-Horizon Tasks**
>
> We completely agree that evaluating on harder, agentic search tasks is valuable. As detailed in our general response (and specifically in our response to **Reviewer 4P4d**), we have conducted new experiments on **BrowseComp, XBench, and BrowseComp-Plus**.
>
> | Model | BrowseComp | XBench | BrowseComp-Plus |
> | :--- | :--- | :--- | :--- |
> | Search-R1 (32B) | 0.5 | 9.3 | 5.1 |
> | **Search-R2 (32B)** | **0.9** | **13.4** | **8.3** |
>
> These datasets require complex, long-horizon search over noisy, open-web environments. Search-R2 consistently and significantly outperforms Search-R1 in these extreme scenarios (e.g., **+80% relative gain on BrowseComp, +44% on XBench**). Notably, our 32B model achieves performance comparable to massive proprietary models like GPT-4o on the Kaggle BrowseComp leaderboard. Please refer to our response to Reviewer 4P4d for the full results table.
>
> ---
>
> Thank you again for your thoughtful review to improve our work. We would be ready to further clarify if any point remains unclear.

---

> > ### Author Rebuttal · Reviewer_WdB7 · 2026-04-03
> >
> > Thank the authors for their comprehensive and thoughtful response. I will keep my score.

---

> > > ### Author Response · Authors · 2026-04-03
> > >
> > > Thank you very much for your kind acknowledgement and for taking the time to carefully review our rebuttal. We are glad to hear that our responses have adequately addressed your concerns.
> > >
> > > We also sincerely appreciate your constructive feedback throughout the review process, which has helped improve the quality and clarity of our work.
> > >
> > > Thank you again for your support.

---

### Decision · Program_Chairs · 2026-04-30

**Decision:**

Accept (regular)

**Comment:**

### Reviews and discussion

Scores after rebuttal: 4, 4, 2 (originally 4, 3, 2).

Reviewer WdB7 (4, conf 4)
- found the paper well-written with clear methodology. Minor concerns about multi-agent framing and memory analysis were resolved during the rebuttal. Score unchanged.

Reviewer NDLU (3 raised to 4, conf 4)
- initially gave a Weak Reject, citing hindsight bias in the process reward (the LLM judge sees the ground-truth answer), reliance on GPT-5.1 as the sole judge, and a narrow evaluation setting. The authors responded with (a) a multi-judge evaluation using Claude Opus 4.6 and 5 human annotators, achieving inter-annotator agreement of kappa 0.68-0.71; (b) scaled baseline comparisons across all model sizes; and (c) an answer-agnostic reward ablation showing only a 0.5-point drop (39.9 vs 40.4 on 7B). The last point was what NDLU had asked for as a condition for raising their score. In their Final Justification, NDLU wrote that "the rebuttal changed my evaluation meaningfully" and raised to Weak Accept.

Reviewer 4P4d (2, conf 5)
- raised two concerns: that Search-R1's reported numbers are lower than in other work (explained by the use of GRPO vs PPO), and that absolute performance on harder benchmarks is low. The authors provided PPO experiments showing Search-R2 improves under both algorithms and BrowseComp/XBench results with +80%/+44% relative gains. In their Final Justification, 4P4d maintained reject, stating that 0.9% on BrowseComp is low compared to WebSailor and MiroThinker, and that those systems "only use web search and browse tools, similar to the retriever used in this work."

### Assessment

I recommend weak accept.

The core contribution, using an Actor-Refiner decomposition with cut-and-regenerate to improve credit assignment in search RL, is clean and practical. The empirical results are consistent across benchmarks, model scales, and RL algorithms, and the overhead is low. The authors did strong rebuttal work, including running entirely new experiments (answer-agnostic reward, PPO comparison, human evaluation with 5 experts) that moved NDLU from Weak Reject to Weak Accept.

The paper's main limitation is that its evaluation uses a controlled retrieval setting (E5 over 2018 Wikipedia). The BrowseComp and XBench results show the method generalizes to harder, noisier settings, but absolute numbers remain modest. The process reward still relies on reference answers during training, though the ablation shows this is not the primary driver of the gains. The theoretical analysis provides useful intuition but falls short of formal guarantees.

Regarding Reviewer 4P4d: their Final Justification rests on comparing Search-R2's absolute BrowseComp score to MiroThinker and WebSailor, which they describe as systems that "only use web search and browse tools, similar to the retriever used in this work." Looking at the cited papers, this does not hold up. Both MiroThinker and WebSailor are full agentic systems with code-execution sandboxes, file management, and live Google Search APIs with open internet access. Search-R2 uses a static E5 retriever over an offline Wikipedia dump. Comparing absolute scores across these two setups does not make sense, and the reviewer's framing of them as equivalent is incorrect. Additionally, 4P4d's Final Justification does not address the PPO experiments that the authors ran specifically in response to their earlier concern. I have given this review reduced weight accordingly.

Two of three reviewers recommend acceptance. The one dissenting review relies on a comparison that conflates the training method contribution with system-level engineering in a different evaluation regime. The paper makes a solid contribution to search-integrated RL and is likely to be useful to practitioners working on retrieval-augmented reasoning.